# Haplotype-aware modeling of cis-regulatory effects highlights the gaps remaining in eQTL data

Nava Ehsan [1], Bence M. Kotis[1], Stephane E. Castel[2,3], Eric J. Song[1], Nicholas Mancuso [4] & Pejman Mohammadi [1,5,6,7] ✉

Expression Quantitative Trait Loci (eQTLs) are critical to understanding the mechanisms underlying disease-associated genomic loci. Nearly all protein-coding genes in the human genome have been associated with one or more eQTLs. Here we introduce a multi-variant generalization of allelic Fold Change (aFC), aFC-n, to enable quantification of the *cis*-regulatory effects in multi-eQTL genes under the assumption that all eQTLs are known and conditionally independent. Applying aFC-n to 458,465 eQTLs in the Genotype-Tissue Expression (GTEx) project data, we demonstrate significant improvements in accuracy over the original model in estimating the eQTL effect sizes and in predicting genetically regulated gene expression over the current tools. We characterize some of the empirical properties of the eQTL data and use this framework to assess the current state of eQTL data in terms of characterizing *cis*-regulatory landscape in individual genomes. Notably, we show that 77.4% of the genes with an allelic imbalance in a sample show 0.5 $\log_2$ fold or more of residual imbalance after accounting for the eQTL data underlining the remaining gap in characterizing regulatory landscape in individual genomes. We further contrast this gap across tissue types, and ancestry backgrounds to identify its correlates and guide future studies.

Genetic variation in the regulatory genome plays a major role in human phenotypic variability and disease susceptibility[1]. Large-scale expression quantitative trait loci (eQTL) mapping efforts in the past decade have identified thousands of common regulatory variants in the human genome that affect gene regulation[2–7]. These data are instrumental for understanding dosage-driven sources of phenotypic variation across individuals and interpreting the statistical signals from trait associated single nucleotide polymorphisms (SNPs) in genome-wide association studies (GWAS)[8–12]. For a given eQTL variant the regulatory effect size can be measured by allelic fold change (aFC), which is the fold difference between the expression of haplotypes carrying the

reference and the alternative allele[13]. The aFC estimates quantify genetic effects on gene expression in an intuitive way and that is consistent with effect sizes from other assays such as allele-specific expression analysis, differential expression analysis, and RT-qPCR, and is mechanistically consistent with *cis*-regulation. Besides biological interpretability, aFC estimates have several mathematically convenient properties that facilitate downstream analysis, and as such, are used in a wide range of applications[3,4,11,14–19].

With the increasing sample size of eQTL transcriptome profiling studies, independent *cis*-eQTL mapping strategies have been developed to identify multiple eQTL signals for each gene in a

[1]Department of Integrative Structural and Computational Biology, The Scripps Research Institute, La Jolla, CA, USA. [2]Department of Systems Biology, Columbia University, New York, NY, USA. [3]New York Genome Center, New York, NY, USA. [4]Center for Genetic Epidemiology, Department of Population and Public Health Sciences, Keck School of Medicine, University of Southern, California, CA, USA. [5]Center for Immunity and Immunotherapies, Seattle Children's Research Institute, Seattle, WA, USA. [6]Department of Pediatrics, University of Washington School of Medicine, Seattle, WA, USA. [7]Department of Genome Sciences, University of Washington, Seattle, WA, USA. ✉e-mail: pejmanm@uw.edu

population[3,20–22]. Notably, the Genotype-Tissue Expression (GTEx) consortium recently used a stepwise regression strategy to map conditionally independent *cis*-eQTL signals in 15,201 RNA-sequencing samples of 838 post-mortem donors across 49 tissue sites[3]. This analysis demonstrated that virtually all protein-coding genes are affected by common genetic regulatory variants with a considerable level of allelic heterogeneity (Fig. 1A). With larger eQTL studies already underway, it is expected that independent *cis*-eQTL signals will be mapped for an increasing number of genes[7,23,24] (Supplementary Fig. 1). However, there are currently no methods available for estimating the aFC effect sizes for multiple independent eQTLs.

Here we introduce a multi-eQTL generalization of the aFC method, aFC-n, for estimating regulatory effect sizes from independent eQTL mapping studies. We benchmark the effect size estimates by aFC-n against those used in the GTEx v8 release[3,13,14] and

characterize their empirical properties and biological correlates. We assess the completeness of eQTL data in terms of characterizing *cis*-regulatory landscape in individual genomes and contrast it across tissue types, and ancestry backgrounds to identify its correlates and guide future studies. Finally, we provide tools and resources to estimate effect sizes and impute gene and haplotype-specific expression using conditional eQTL data.

## Results

### Multi-variant generalization of allelic effects

Under the aFC model, the expression associated with the reference ($e_R$) and the alternative ($e_A$) eQTL alleles in an individual are determined by a shared basal gene expression, $e_B$, and allele-specific regulatory activities, $k_R$, and $k_A$ such that $e_R = e_B k_R$, and $e_A = e_B k_A$. The total gene expression in an individual is the sum of the two allele-specific

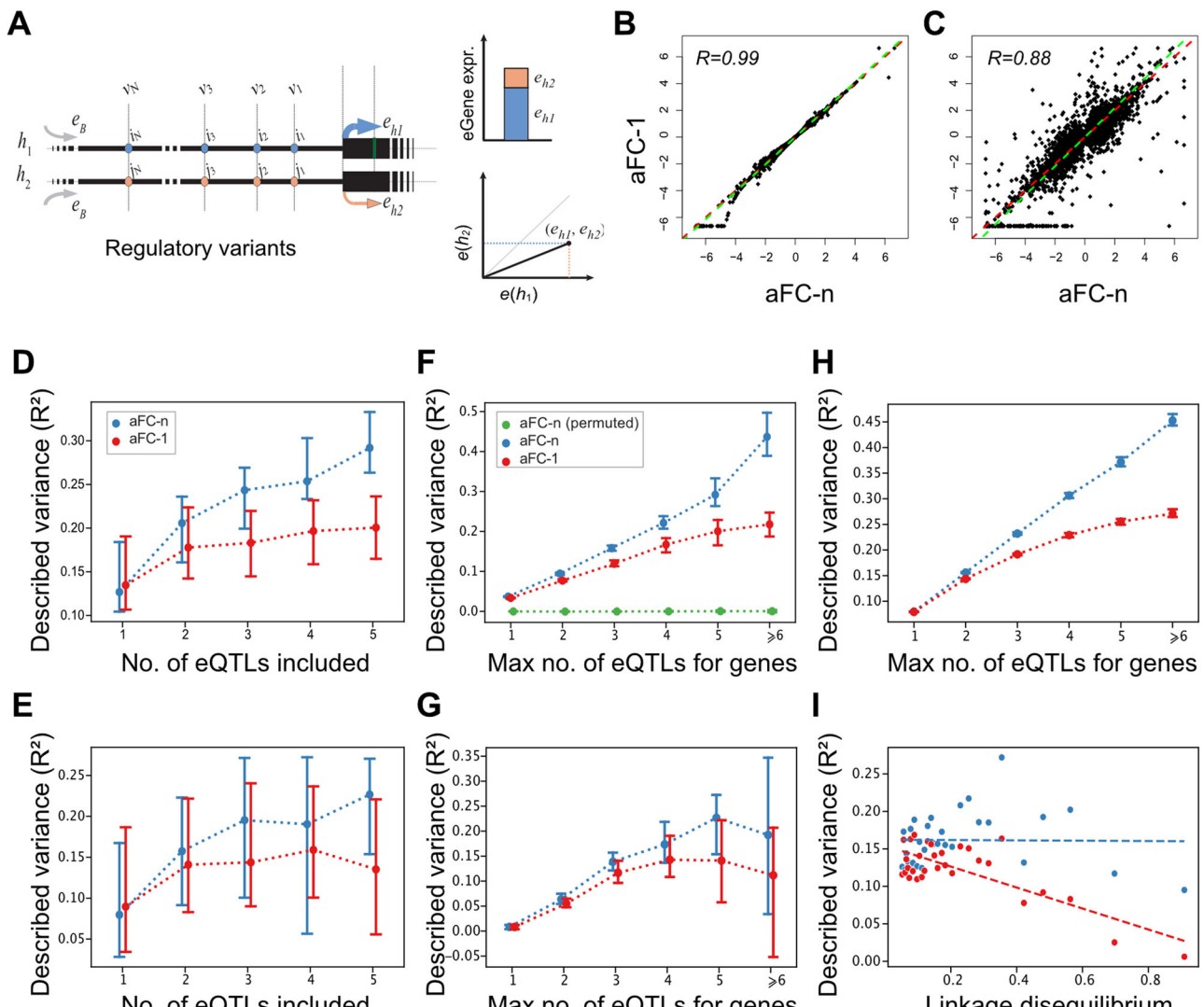

**Fig. 1 | aFC-n improves the accuracy of eQTL effect size estimates compared to aFC-1. A** Schematic representation of aFC-n with N *cis*-eQTLs ($v_1,...,v_N$). Haplotypes $h_1$ and $h_2$, share basal expression level $e_B$, and $e_{h1}$ and $e_{h2}$ are the haplotypic expression terms. The effect size estimates from aFC-1 and aFC-n for adipose subcutaneous tissue, for genes with a single ((**B**); $n = 8509$), and multiple ((**C**); $n = 17,166$) eQTLs. $R$ denotes Pearson correlation coefficient, Deming regression fit is shown in green, and the red line is ($y = x$). Prediction accuracy for gene expression ((**D**); $n = 103$), and allelic imbalance ((**E**); $n = 80$) in genes with five eQTLs as a function of the number of eQTLs considered. eQTLs are included in the order they were mapped. Variance explained by eQTLs in gene expression ((**F**); n = 10,358) and

allelic imbalance ((**G**); $n = 8495$). The number of genes varies across different bins. **H** Variance explained by eQTLs in gene expression aggregated across 49 GTEx tissues ($n = 462,449$). **I** The prediction accuracy for genes associated with two independent eQTLs as a function of linkage disequilibrium (LD). The regulatory variants are filtered to address eQTLs with LD ≥ 0.05 (24,930 variant-pair). The x-axis represents the median LD, among 30 equally sized bins and the y-axis represents the median $R^2$ for the genes within each bin. The n value denotes the number of genes. In (**D, H**) error bars represent bootstrap 95% confidence intervals of the median.

expressions. The regulatory effect size, aFC, is $\delta_{A,R} = \frac{k_A}{k_R}$, which for a single eQTL, can be calculated using the previously published aFC quantification tool, hereafter referred to as *aFC-1*[13]. Here, we introduce *aFC-n* generalizing the model to a haplotype with $N$ independent eQTLs. The allele-specific expression in aFC-n is $e_{i_1 \ldots i_N} = e_R \prod_{n=1}^{N} \delta_{i_n,R}^{(n)}$, where $i_n$ and $\delta_{i_n,R}^{(n)}$ are the present allele, and the associated aFC for the $n^{\text{th}}$ eQTL, respectively, and $e_R$ is the expression of a haplotype carrying reference allele for all eQTLs. We infer the maximum likelihood parameters for this model under log-normal assumption to estimate aFC associated with all independent eQTLs affecting a gene using phased genotypes and gene expression counts ("Methods").

### The aFC-n improves the accuracy of the *cis*-regulatory effect size estimates

To validate aFC-n, we used the empirical distribution of the adipose subcutaneous tissue in GTEx v8 eQTL data to simulate genetic regulatory effects in 15,167 genes with 1 to 14 eQTLs (Methods). In this simulation study, aFC-n consistently estimated the effect size accurately across all genes, when all eQTL variants of a gene were included in the model (Supplementary Fig. 2). Applying aFC-n to GTEx project data v8, we estimated regulatory effect sizes for a total of 458,465 conditionally independent eQTLs from 49 tissues (Supplementary Data 1; Supplementary Fig. 3). As expected, the effect size estimates from aFC-n were well correlated with the current effect size estimates from GTEx v8 eQTLs that were calculated using aFC-1. The correlation ranges from 89 to 98% across tissues for genes with a single eQTL where the two methods are mathematically identical, and from 80%-92% for genes with multiple eQTLs (Fig. 1B, C). We used adipose subcutaneous tissue data to compare the accuracy of the aFC estimates from the two methods in predicting gene expression and in predicting allele-specific expression across GTEx individuals ("Methods"). Since both methods are agnostic to allele-specific expression data, using allelic imbalance prediction accuracy allows us to evaluate the quality of the effect size estimates in an orthogonal way[13]. We compared the predicted gene expressions from each set of effect size estimates with the observed gene expression after log-transformation and PEER correction[3]. The predicted allelic imbalance was benchmarked against the observed logit-transformed haplotype-aggregated allelic expression generated by phASER[14,25]. Using genes that each have five conditionally independent eQTLs, we predicted gene and allelic expression five times each time including the effect size of one additional eQTL in the prediction. For the effect size estimates from aFC-1, we observed that including additional eQTLs leads to limited improvement in gene expression prediction accuracy and no increase in the prediction accuracy for allelic imbalance beyond what is achievable by accounting for the top eQTL genotype only. In contrast, the new effect size estimates by aFC-n delivered progressively better predictions as more eQTLs were included in the prediction of both gene and allelic expression (Fig. 1D, E). Next, we used all genes with eQTLs to compare the overall prediction accuracy when all known eQTLs are considered for each gene. The accuracy gap between the predictions from aFC-n and aFC-1 was widened progressively in genes with more known eQTLs and overall the predictions were significantly more accurate for multi-eQTL genes (ranksum test p-value $3.28 \times 10^{-27}$ for gene expression, and $5.07 \times 10^{-8}$ for allelic imbalance) (Fig. 1F, G). We obtained a similar pattern for total expression prediction in 49 GTEx tissues (Fig. 1H; Supplementary Fig. 4; Supplementary Fig. 5).

Different conditionally independent eQTLs for a gene are regularly in linkage disequilibrium (LD). To further explore the effect of linkage disequilibrium (LD) on the performance of the aFC-1 and aFC-n, we simulated effect sizes for 2-eQTL genes with LD ranging from 0 to 0.9 ("Methods"). When eQTL variants within a gene are correlated, aFC-1 would return biased estimates which leads to a lower

performance for effect size estimation and prediction accuracy in higher LD, while aFC-n showed consistent performance at all LD values (Supplementary Fig. 6A, B). Moreover, for 2-eQTL genes in GTEx data, the prediction accuracy gap between aFC-n and aFC-1 was widened as eQTLs of a gene are in higher LD (Fig. 1I; Supplementary Fig. 6C), indicating that considering the regulatory effects of all eQTLs simultaneously, as is done in aFC-n, is critical for accurate estimation of regulatory effect in presence of linkage disequilibrium between eQTLs.

### Empirical properties of the estimated effect sizes

Next, we used aFC estimates from adipose subcutaneous tissue to characterize the regulatory effects of the independent eQTLs identified in GTEx project data. We found that 15.2% of the 25,675 independent eQTLs identified in adipose tissue altered the expression of a haplotype by more than twofold (Supplementary Fig. 7A). In addition, across all genes, secondary eQTLs (eQTLs were ranked with the order they were mapped in stepwise regression[3]) tended to have lower minor allele frequencies (Supplementary Fig. 7B) and larger regulatory effects (Fig. 2A, B). However, we found that the eQTLs in genes with many eQTLs tended to be stronger in general (Fig. 2C). Accounting for this, we found that for a given gene the relative effect size of the eQTLs with respect to the average eQTL decreased with the order they were mapped (Fig. 2D).

Next, we compared the *cis*-regulatory effects in eQTL and ASE data. We found that aFC effect sizes estimated from eQTL data were highly consistent with the median observed allelic imbalance among individuals that are heterozygous for the top eQTL, with sufficient >10 heterozygous individuals and minimum read coverage 8, (rank corr. 0.76±0.01, Deming regression slope 0.9; Fig. 2E). We further found that the concordance with the ASE data was decreased for secondary eQTLs (Fig. 2F) partly due to the decreased minor allele frequency (Fig. 2A; Supplementary Fig. 7B), and partly due to the drop in haplotype phasing accuracy[14,25].

### The aFC-n improves the prediction of genetically regulated gene expression

Next, we sought to demonstrate the application of aFC-n in predicting gene expression levels. Genetically driven gene expression has been widely used to identify transcriptome-mediated association signals in complex traits[10,26]. We used GTEx v6p data from 316 adipose samples to build a predictive model using 4696 conditionally independent eQTLs spanning over 3970 protein coding genes ("Methods") and evaluated the performance on 265 unseen samples exclusive to GTEx v8 release. For predicting expression using the aFC model we used independent eQTLs derived from GTEx v6p data (mean 1.2 eQTLs per gene). We used elastic net (enet)[27] and Sum-of-Single-Effects (SuSiE)[28], two powerful and robust methods used for predicting gene expression from genetic data to benchmark the accuracy of predicted gene expressions from eQTL effect sizes ("Methods") and for that, we used (1) all genetic variants in the 1 Mb window around each gene meeting our QC criteria (Fig. 3, Supplementary Fig. 8A, B), and (2) conditionally independent eQTLs for a gene (Supplementary Fig. 8C, D). Restricting the comparison to genes with *cis*-heritability $p$ value < 0.01 present in eQTL data, we found that the prediction performance of the eQTL genotypes in the aFC model was higher than the two state-of-the-art methods in unseen samples.

Consistent with previous reports[29–31], we observed a notable reduction in prediction accuracy among African American ancestry individuals in all three models (Supplementary Fig. 9A). To alleviate this issue, we devised a hierarchical extension of the aFC-n model to allow ancestry-specific aFC estimates when supported by data (Methods). We found that the ancestry-specific signals identified by this model were generally false positives driven by low sample sizes and therefore failing to diminish the performance gap between different ancestry groups (Supplementary Fig. 9B).

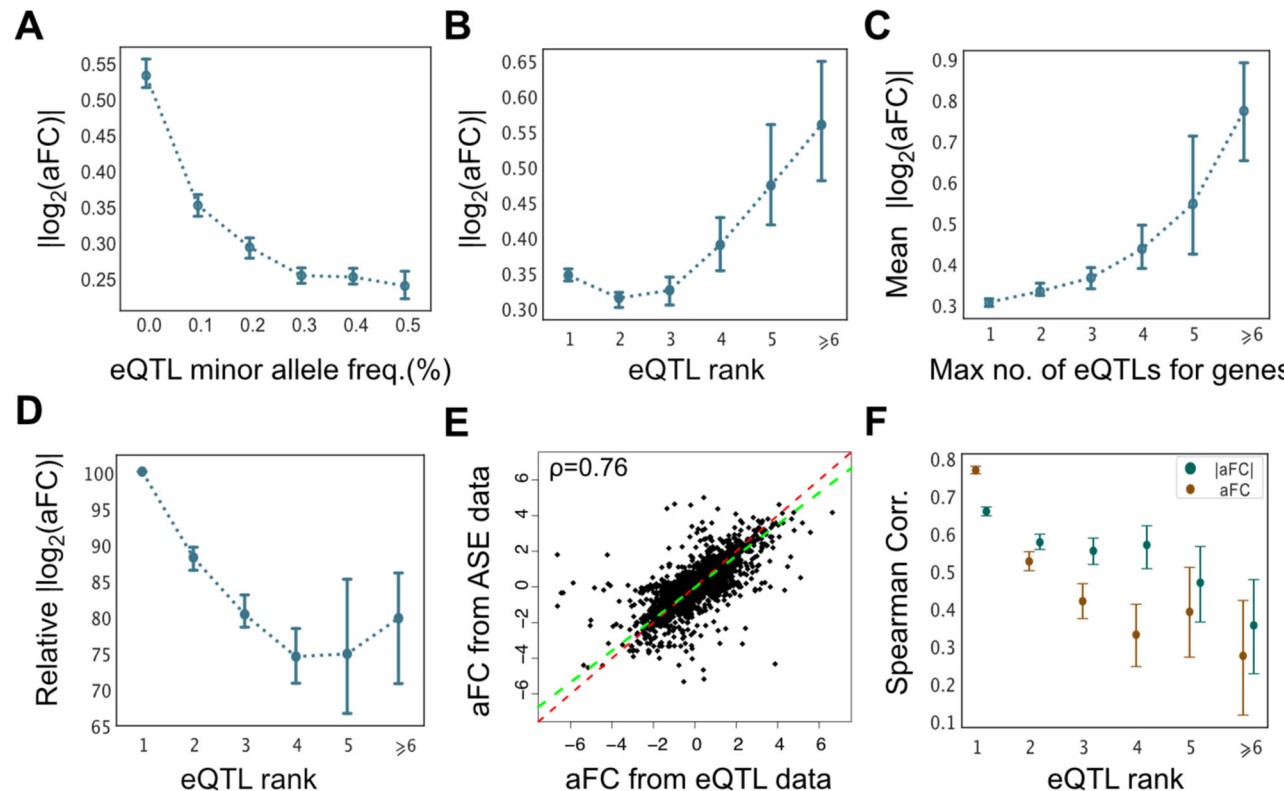

**Fig. 2 | Empirical properties of conditionally independent eQTL effect sizes.** The absolute effect size as a function of eQTL minor allele frequency (**A**) and the order at which it was identified for a gene (eQTL rank) (**B**) in adipose subcutaneous tissue. **C** Average eQTL effect size as a function of the total number of eQTLs associated with a gene. **D** The eQTL effect size relative to the average eQTL effect for a given gene as a function of the eQTL rank. (**A**–**D** $n = 25,682$.) **E** The eQTL effect sizes as estimated by gene expression and population ASE data independently. The Deming regression fit is shown in green, and the red line is ($y = x$). **F** The correlation with the ASE data progressively decreased for secondary eQTLs ($n = 25,682$). Error bars represent 95% bootstrap confidence intervals of the median. The n value denotes the number of genes.

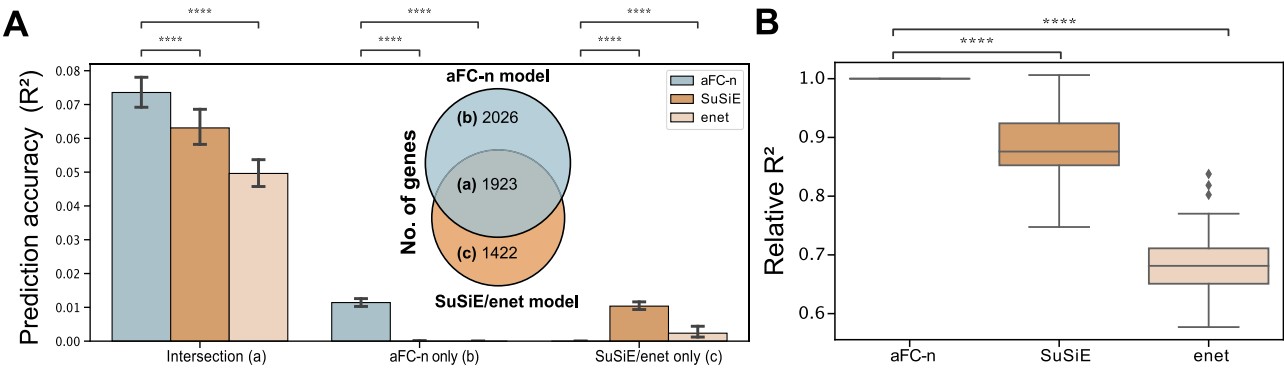

**Fig. 3 | The aFC-n improves the predictive accuracy of genetically regulated gene expression.** Comparing predicted gene expression from the aFC-n, elastic net (enet)[27] and SuSiE[28] models, using out of sample data. For SuSiE and enet we considered genes with significant expression *cis*-heritability. For aFC-n we considered the model for genes with eQTLs. **A** The aFC-n model outperforms SuSiE and elastic net for 1923 genes shared by all models (subpanel a) in adipose subcutaneous tissue (median $R^2$ is 0.073 for the aFC-n model and 0.063 for the SuSiE predictive model; The two-sided Wilcoxon signed-rank test $p$ value: $1.8 \times 10^{-47}$). For the genes not shared between models, the performance is much lower for all models (subpanels b, c). Error bars represent 95% bootstrap confidence intervals of the median. **B** The distribution of the median $R^2$ relative to the median of the $R^2$ for aFC-n model for 47 tissues for shared genes (two-sided wilcoxon signed rank test $p$ value = $2.6 \times 10^{-9}$ and $2.4 \times 10^{-9}$, comparing aFC-n with SuSie and enet, respectively). Comparing aFC-n with SuSiE and elastic net prediction models, Wilcoxon signed-rank tests are significant (FDR < 0.05) for 46 and 47 tissues, respectively. The $p$ value annotation: ****$p \leq 10^{-4}$. Boxplots represent first quartile, median, and third quartiles. Whiskers represent Q1–1.5* interquartile range (IQR) and, Q3 + 1.5*IQR.

## Majority of the observed allelic imbalance in individual samples is not described by the current eQTL data

Next, we sought to determine the fraction of genes where the *cis*-regulatory landscape is adequately characterized at an individual level by their genotype at known eQTLs. We used the observed allelic imbalance in a gene as the ground truth under the rationale that the ASE data is the net regulatory effect of all heterozygous *cis*-acting variants affecting a gene—including those that are not known eQTLs. Furthermore, allelic imbalance is almost entirely driven by genetic factors, with heritability estimates above 85%[32]. Specifically, for each gene in an individual, we checked if the allelic imbalance is consistent with predictions from the eQTL data and identified cases

where we observed excess imbalance beyond 0.5 and 1 aFC ("Methods"). We performed power analysis to account for the confounding effect of limited statistical power in detecting an excess allelic imbalance in low expressed genes (Supplementary Fig. 10). Looking at the adipose subcutaneous tissue samples we found that the number of genes with excess allelic imbalance initially increased but then it dropped progressively with increased statistical power at the right end of the axis (Fig. 4A). The unexpected drop in spite of the high statistical power to detect allelic

imbalance in these genes is due to a general tendency in highly expressed genes to be intolerant of genetic variation (Supplementary Fig. 10D, E)[13,15,33]. We limited our analysis to protein-coding genes expressed >1TPM and >80% statistical power to identify 0.5-fold resolution in $\log_2$ aFC scale. We found that on average between 3.2% (Brain - Frontal Cortex (BA9)) and 8.6% (Liver) of the considered genes across different tissues in an individual showed excess allelic imbalance beyond what is expected from the genotypes at known eQTL variants. This constituted on average a 22.6%

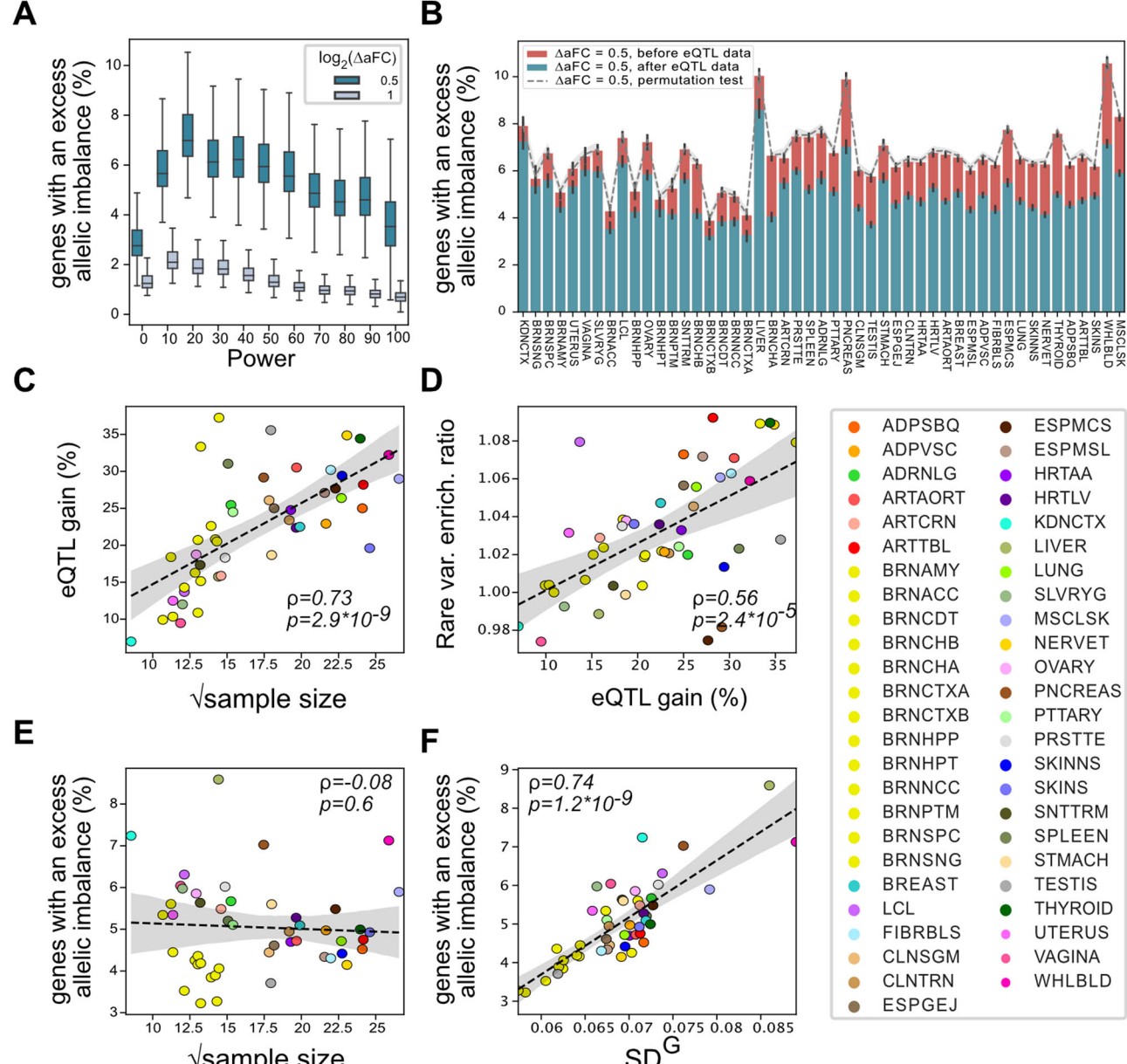

**Fig. 4 | Individual-level *cis*-regulatory landscape captured by eQTL data.** The percentage of genes with an excess allelic imbalance after accounting for known eQTLs as a function of power in adipose subcutaneous (**A**) (see Supplementary Data 2 for detailed statistics for each individual), and for genes with over 80% power across all tissue samples (**B**). Panel (**B**) shows the fraction of genes with half a fold excess allelic imbalance with the red bars representing the baseline case where no eQTL data is considered (see Supplementary Data 1, and 3 for tissue names and the number of samples/genes per tissue, respectively). The Error bars represent bootstrap 95% confidence intervals. **C** The relative decrease in the number of genes with excess allelic imbalance after incorporating eQTL data (eQTL gain) as a function of the tissue sample size. **D** Enrichment of rare variants (5% FDR) increased

with the eQTL gain across tissues. **E, F** The proportion of genes with excess allelic imbalance after incorporating eQTL data (green bars in **B**) did not show correlation with the tissue sample size (**E**) but was highly correlated with the median expected genetic variation in gene expression in a given tissue. $SD^G = \sqrt{V^G}$ where $V^G$ is estimated by ANEVA[15]. For panels (**C–F**), linear regression fit is shown in black dashed line along with bootstrap 95% confidence intervals in gray shades. The $\rho$ and $p$ values represent the two-sided spearman correlation coefficient and associated p-value, respectively. Boxplot in (**A**) represent first quartile, median, and third quartiles. Whiskers represent Q1−1.5* interquartile range (IQR) and, Q3 + 1.5*IQR. Outliers are hidden for ease of viewing.

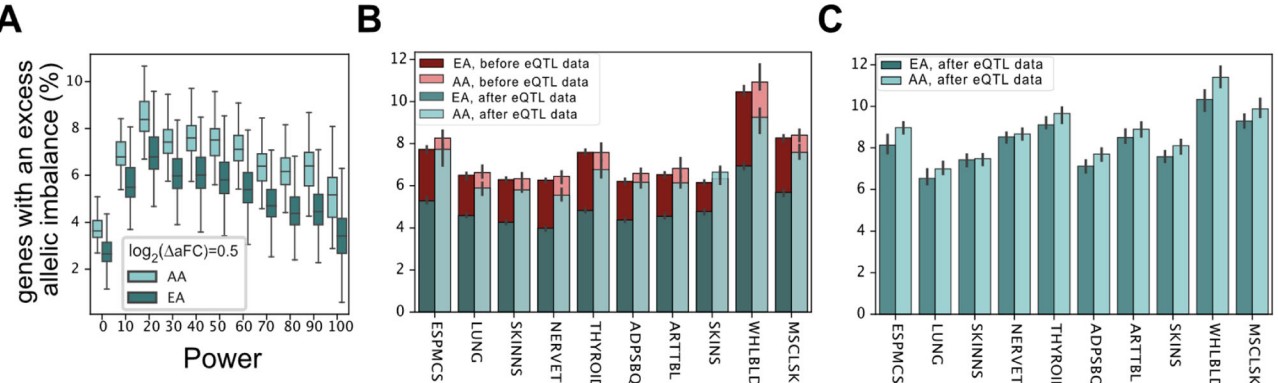

**Fig. 5 | Self-reported ancestry, and the accuracy gap in capturing individual-level *cis*-regulatory landscape by eQTL data.** The percentage of genes with an excess allelic imbalance by self-reported ancestry after accounting for known eQTLs as a function of power in adipose subcutaneous (**A**) (see Supplementary Data 4 for detailed statistics for each individual), and for genes with over 80% power across ten GTEx tissues with largest sample sizes (**B**, **C**). AA and EA denote African American, and European American individuals, respectively. The eQTL data used in panels (**A**, **B**) is from the official GTEx release, while for panel (**C**) the eQTLs were mapped using a subset of the samples with equal number of AA and EA individuals. Error bars in (**B**, **C**) represent 95% bootstrap confidence intervals of the median (see Supplementary Data 5 and Supplementary Data 6 for number of analyzed samples/genes per tissue/population). Boxplots in panel (**A**) represent first quartile, median, and third quartiles. Whiskers represent Q1–1.5* interquartile range (IQR) and, Q3 + 1.5*IQR. Outliers are hidden for ease of viewing.

decrease from a baseline scenario where no eQTL knowledge was available and both haplotypes were expected to be expressed equally in all genes (Fig. 4B). This decrease represents the gene regulatory knowledge *gained* by the GTEx consortium eQTL analysis. As expected, the gain was highly correlated to the sample size of the tissues as more *cis*-regulatory variants were identified (Fig. 4C; Supplementary Fig. 11). The enrichment of rare variants among genes with excess allelic imbalance (5% FDR) also increased with the gain across tissues, confirming that with increased statistical power the eQTL model better captures the common variant regulatory effects and the excess allelic effects by rare variants (Fig. 4D). However, we found that in contrast to the relative gain, the absolute fraction of genes with an excess allelic imbalance in each tissue was not correlated with the sample size used in the eQTL analysis (Fig. 4E) but instead was correlated with the median amount of heritable variation in gene expression ($V^G$) in each tissue as estimated by the analysis of expression variation (ANEVA)[15] (Fig. 4F).

**Matching sample counts alone will not close the accuracy gap in African ancestry individuals**

GTEx data includes mostly individuals of European descent[3]. To assess how well the *cis*-regulatory landscape of the genes is captured across different ancestry backgrounds, we further stratified the analysis by self-reported ancestry for each GTEx donor. Looking at the adipose subcutaneous tissue samples, we found that the percentage of genes with excess allelic imbalance was significantly higher among African Americans suggesting that the eQTL data does not capture the *cis*-regulatory landscape in African ancestry individuals as accurately (ranksum test *p* value = 2.3 × 10⁻²⁵ for genes with 80% power; Fig. 5A). Looking at the ten most sampled tissues, we observed that the gain due to eQTL data is systematically lower in the African ancestry individuals in line with the lower sample sizes (Fig. 5B). To exclude the effect of sample size, we repeated the eQTL mapping using the same number of European and African ancestry individuals for each tissue. We found that the number of genes with excess allelic imbalance was still higher in the African ancestry individuals (Wilcoxon signed-rank test, *p* value = 0.002; Fig. 5C). While the number of samples used in the eQTL analysis was identical for both ancestry groups the remaining gap is likely due to a higher level of regulatory variation and/or reference bias in ASE data in the African American population (Fig. 5C; Supplementary Fig. 12) highlighting additional obstacles that impede analysis of non-European ancestry genomes.

## Discussion

With the increasing size of the eQTL studies, many genes are associated with multiple eQTL variants. Here we introduced a new multi-variant method, aFC-n, to estimate the *cis*-regulatory effect size for multiple independent eQTLs associated with a gene. Applying aFC-n to GTEx v8 project data we showed that the resulting eQTL effect sizes are significantly more accurate than the currently available estimates in predicting the *cis*-genetic effect on expression, and ASE data. Using these effect sizes to predict allelic imbalance, we showed that the current eQTL data is highly consistent with the observed allelic imbalance at the population level.

We showed that aFC-n provides accurate estimates of cis-regulatory effects when all regulatory variants affecting a gene are known and conditionally independent. Violation of these assumptions can affect the quality of the results. Specifically, we demonstrated how the LD between two eQTLs affecting the same gene can systematically erode the performance of the aFC-1. While this experiment demonstrates the strength of aFC-n by simultaneously estimating all multiple effect sizes, it also highlights its limitation in cases where one or more eQTLs are not included in the model. Specifically, when a gene is affected by other eQTL beyond what is included in the model, the effect size estimates will be systematically biased by the contaminating effect of the LD. Furthermore, aFC-n assumes that eQTL genotypes are linearly independent which is a critical condition for mathematical identifiability of the effect sizes and as such inherently satisfied in conditional eQTL datasets. However, the general application of aFC-n model on an arbitrary set of SNPs will require the addition of appropriate shrinkage penalties to enable parameter inference. Moreover, aFC-n assumes biologically independent among eQTLs in that it does not allow for epistatic interactions. While there are many biological scenarios under which two regulatory variants can have nonadditive effects, we have previously shown that this assumption is rarely violated for the eQTL variants identified by stepwise regression approach[3].

The extent of regulatory variation represented by eQTL data has been previously explored implicitly by heritability analysis[32], and by quantifying the diminishing number of identified eQTLs at a certain sample size[3,4]. A comprehensive catalog of the eQTLs is critical for identifying dosage-driven phenotypic variation and GWAS interpretation[8,9,11]. Here we employed the aFC framework to explicitly assess the eQTL data in representing individual-level regulatory landscape using ASE data. We found that the current

eQTL data from GTEx adequately represented the net *cis*-regulatory effects on the majority of the gene haplotypes across tissues. However, this is mainly due to the fact that most genes in a given sample do not show significant allelic imbalance. Controlling for the statistical power and at half a $\log_2$ fold resolution, we found that just over a fifth of the genes with allelic imbalance in a sample is accounted by the current eQTL data; indicating that a majority of *cis*-regulatory genetic variants, likely with low minor allele frequencies, are yet to be mapped. As expected, and in line with the increased power in eQTL mapping, the tissue types with larger sample sizes in GTEx data tended to show a higher gain in describing genes with allelic imbalance. However, the amount of residual regulatory variation across samples of a given tissue type was not correlated with the sample size, instead it was strongly correlated with the total amount of genetic regulatory variation in a given tissue as estimated by ANEVA. Considering that tissue samples are from the same individuals, it is implausible that this observation would be a technical artifact driven by reference bias in ASE data. We postulate that the adequate sample size for an eQTL study is not a fixed number and ultimately depends on the specific biological context being studied.

A similar analysis across European and African American individuals defined by self-reported ancestry demonstrated that the GTEx eQTL data offers lower gains in describing allelic imbalance in African American individuals and is overall less informative. This general issue is well recognized in the field and is in line with the fact that most individuals included in the GTEx project are of European ancestry[3,34,35]. However, we found that the performance gap between European and African Americans decreased but did not entirely disappear even when the sample sizes were matched. This observation was consistent with the higher amount of variation in ASE data as measured by ANEVA and in line with the extensive genomic diversity in Africa. However, unlike the cross-tissue analysis, we cannot rule out a contribution of technical artifacts due to reference bias. Nevertheless, the de facto performance gap, which goes beyond the sample size effects, suggest that reaching the same predictive accuracy in analyzing African American genomes will likely entail not only improvement in methods and references but also the inclusion of a relatively higher number of samples to enable characterization of an inherently more diverse and admixed population.

Finally, while haplotype-aware eQTL mapping methods have been proposed[36–38], genome phasing quality remains a bottleneck in capturing the regulatory landscape in human haplotypes. We expect long-range phasing of the genomes beyond what is currently achievable using short-read sequencing[25] to be a valuable addition to the reference transcriptomic cohorts such as GTEx, and TOPMed[23]. Transcriptome-wide association studies utilize genetically predicted gene expression to identify phenotypic variations that are likely mediated through genetically driven gene dosage. Our method is distinct from the current approaches in that it uses a mechanistic method to predict gene expression in a haplotype-specific fashion using a relatively small set of known eQTL variants. We showed that this model of genetic variation in *cis*-regulation yielded higher predictive accuracy than the state-of-the-art gene expression prediction methods.

Unlike the conventional regression-based methods, predictions of haplotype-level dosage cannot be derived and used to analyze associations based on summary statistics. However, others have developed approximation methods that utilize aFC framework for gene dosage prediction and related association analyses using summary statistics[38]. Increased availability of large-scale genomic datasets on cloud services such as the UKBB research analysis platform and the NHGRI genomic data science analysis, visualization, and informatics lab-space (AnVIL) will facilitate individual-level analysis of genetic data in the coming years.

## Methods

### Haplotypic aFC estimation

The expression associated with eQTL alleles on each haplotype is described with a shared basal gene expression, $e_B$ and allele-specific factors $k_R$, and $k_A$. The regulatory effect size, aFC, is defined as $\delta_{A,R} = \frac{k_A}{k_R}$. Considering the case of $N$ eQTLs acting on the same gene independently and defining $s_{A,R} = \log_2 \delta_{A,R}$, the expression of the haplotype carrying $N$ variants is:

$$\log_2 e_{i_1 \ldots i_N} = \log_2 e_R + \sum_{n=1}^{N} s_{i_n,R}^{(n)} \quad (1)$$

where $s_{i_n,R}^{(n)}$ is the log aFC associated with the allele $i_n$ of the $n^{\text{th}}$ eQTL, and $e_R$ is the expression of a haplotype carrying reference allele. This generalized aFC model is used in a variance stabilized non-linear regression to estimate aFCs associated with all independent eQTL variants affecting a given gene, simultaneously. Assuming a multiplicative noise model[13], the haplotype aware aFC could be estimated as the least-squares solution to the following nonlinear equation using a log-normal noise assumption:

$$\log_2 e_{<i_1 \ldots i_N>,<j_1 \ldots j_N>} = \log_2 \left( 2^{s.h1} + 2^{s.h2} \right) + \log_2 e_R \quad (2)$$

where $e_{<i_1 \ldots i_N>,<j_1 \ldots j_N>}$ is the total gene expression which is the sum of the two haplotypic counts, and $h_1$ and $h_2$ are binary indicator vectors representing the phased genotype of each allele, and $s$ is the vector denoting the log-transformed effect sizes of eQTLs.

To estimate the model parameters, we used gene expression read counts. We normalized the counts for library size, added 1 pseudo-count and log-transformed to stabilize the variance for least-square fitting. The expressions were corrected for significant linear effects of identified confounding factors using PEER[39], top 5 genotype-based principal components, sequencing platform (Illumina HiSeq 2000 or HiSeq X), sequencing protocol (PCR-based or free) and sex. The correction was done in two steps: first, we regressed the expression vector of each gene against covariates and selected those with nominally significant coefficients ($p < 0.01$). Then we regressed the expression vector on selected covariates and set the residuals as the corrected expression vector which was used for effect size calculation[13].

The log aFCs for eQTLs were calculated using non-linear least-squares regression and were constrained to $\pm \log_2(100)$ to avoid outliers. The initialization step is a linear fit where the haplotype vector was used as an independent variable and the adjusted expression was the dependent variable. The coefficients were used as the initial values of the vector $s$ in the nonlinear optimization function. This makes the LM algorithm converge closer to the real value. We used the Python non-linear least-squares minimization and curve fitting (LMFIT) library. We calculated the effect size estimates for GTEx v8 independent eQTLs. Confidence intervals were calculated to infer 95% confidence intervals for the aFC estimates. In GTEx v8 data, the range of those eQTLs whose 95% confidence interval of aFC estimates overlapped zero varied from 11.6% (kidney-cortex) to 39.5% (cells-cultured-fibroblasts) across tissues if PEER correction was not included, and the range narrowed between 0.9% (Brain-Substantia-nigra) and 2.9% (Testis) when correcting for confounding variation for aFC estimates.

Conditional independence assumption used in stepwise regression approach for deriving eQTLs identifies independent signals for gene expression and chooses the variants that describe the best signal while controlling for covariates and all other mapped eQTL signals[3,4]. This enables us to estimate the regulatory effects by ensuring stability and identifiability of the model parameters in the optimization process.

## Simulation scheme

We used simulated data to validate aFC-n implementation and inference stability. We used 15,167 genes from adipose subcutaneous, their associated eQTLs (range: 1 to 14 eQTLs per gene), and individual genotypes for simulation. We synthesized log2 effect sizes for each eQTL from standard Normal distribution ($norm[0,\sigma=1]$). We calculated the expected haplotypic expression using Eq. (1) and used the empirical gene expression average for each gene to set $e_R$. Expected gene expression for each GTEx individual was estimated as the sum of the two haplotypic counts, and the simulated sequencing read count for each gene was generated using a Poisson distribution. To generate simulated ASE data, we calculated the expected reference expression ratio for each individual as $r = \frac{e_{h1}}{e_{h1}+e_{h2}}$, and simulated discrete read counts for each haplotype using Binomial distribution. Notably, we did not include any additional biological variation in our simulated data beyond genotypic differences among the individuals, and the sampling noise associated with the count nature of the data. This was to maintain simplicity since the simulations were intended for validating the code and inference procedure and not for reporting performance measures which we performed using real GTEx data.

## The effect size estimation under different LD levels

To explain the impact of LD structure on effect size estimation, we used GTEx v8 variant calls[3] as the reference panel and we selected pairs of eQTLs in GTEx data with their LD ranging from 0 to 0.9, ($R^2$ calculated with PLINK 2.0 "`plink --bfile reference-panel --ld SNP1 SNP2`"). For each variant we assigned a simulated aFC from a normal distribution $norm[0,\sigma=1]$. That resulted in 2000 pairs of variants (200 for each bin of LD). The simulated expression counts for 839 samples were calculated using Eq. (2) using a Poisson distribution. The expressions were log-transformed and normalized to use for calculating effect size estimates under aFC-n and aFC-1 models.

## Imputing gene and allelic expression

The estimated effect sizes were used to predict the *cis*-genetic effects on gene expression and allelic imbalance in individuals with phased genotype data. The evaluation is assessed against gene expression after log-transformation and PEER correction[3] and haplotype-aggregated allelic expression generated by phASER[14,25]. To smooth the haplotype counts, a pseudo-count of 0.5 was added to each observed haplotype expression and the minimum total coverage to be included in the calculations was set to 100. This makes the data well powered to detect allelic imbalance (Supplementary Fig. 10A; Supplementary Fig. 13A). The log-transformed estimated gene expression and Allelic Imbalance $AI$ for an individual were derived from equation (Eq. (3)),

$$\log_2 e_{<i_1\dots i_N>,<j_1\dots j_N>} = \log_2(2^{s.h1} + 2^{s.h2})$$
$$AI_{<i_1\dots i_N>,<j_1\dots j_N>} = s.h1 - s.h2, \tag{3}$$

Where, $h_1$ and $h_2$ are binary indicator vectors representing the genotype of each allele of the individual and $s$ is the vector denoting the log-transformed effect sizes of eQTLs of interest and $s.h_i$ is the sum of effect sizes for alternative alleles.

To compare the accuracy of *cis*-regulatory variation described by the aFC estimates derived by the two methods (aFC-n and aFC-1), we used the effect sizes of 25,232 and 17,147 eQTL variants in adipose subcutaneous tissue samples from 581 individuals for gene expression and allelic expression, respectively. Due to the constraint on minimum coverage for ASE analysis, the number of individuals taken into account differs for each gene (Supplementary Fig. 13B). The prediction accuracy for predicting gene expression log fold-change, and log-transformed allelic imbalance was measured by coefficient of determination, $R^2$:

$$R^2 = 1 - \frac{\sum_{individuals}(observation - prediction)^2}{\sum_{individuals}(observation)^2} \tag{4}$$

where the prediction values are $\log_2 e_{<i_1\dots i_N>,<j_1\dots j_N>}$ for gene expression, and $AI_{<i_1\dots i_N>,<j_1\dots j_N>}$ for allelic imbalance as provided in Eq. (3), and the observed values are what is measured for gene expression fold change from mean, and log-transformed allelic imbalance measured for each individual in GTEx data. First, the comparison was done based on the number of eQTLs included for each gene. By using the effect sizes derived from aFC-n model (Eq. (2)), the median $R^2$ increased as more variants were taken into account for both gene expression and allelic imbalance (Fig. 1D, E).

Both models performed similarly for genes with one eQTL but the portion of total expression variation that could be explained by the known eQTLs for gene expression and allelic imbalance showed an increasing pattern for the median $R^2$, when genes have more regulatory variants (Fig. 1F, G). With an increased number of eQTLs for a gene, we observed a steadily increasing gap for the prediction accuracy between aFC-n and aFC-1 in multi-eQTL genes (Fig. 1F). We used a permutation test to ensure that the improved performance of aFC-n versus aFC-1 is not driven by overfitting. Specifically, we permuted the individual sample IDs to decouple genotype and gene expression variation while retaining the data size, allele frequencies and LD structure. We found that the $R^2$ for the aFC-n predictions of gene expression in the permuted dataset remains at zero ruling out systematic overfitting to the data (Fig. 1F). To further evaluate the effect sizes on an independent data we used haplotype-aggregated allelic expression since the trained model is agnostic to allele-specific expression (Fig. 1G).

The ASE prediction was compared with the read counts with WASP mapping strategy[36] to reduce the mapping bias that is sometimes present in ASE analysis. On average, WASP correction improved prediction about 7 percent for 9503 genes at a minimum coverage of 100 reads in adipose subcutaneous tissue.

## *Mapping conditionally independent eQTLs in GTEx v6p data*

To perform independent *cis*-eQTL mapping based on GTEx v6p samples, gene expression values from tissue samples were log-transformed and normalized and limited to autosomal genes with more than 5 reads in at least 10 individuals. The *cis*-eQTL mapping is performed using tensorQTL (`--mode cis_independent`)[22], based on the stepwise regression approach described in[3], using WGS-based genotypes and restricting the analysis to genes that have minor allele frequency for all eQTLs above 0.01.

## Benchmarking gene expression prediction models

To compare aFC-n with gene expression prediction models, we fitted eQTL data for 316 individuals with measured gene expression in adipose subcutaneous tissue in the GTEx v6 using an elastic net and the *Sum-of-Single-Effects* model (i.e. SuSiE). For each gene, we focused on local genetic variation flanking 0.5 Mb up/downstream of the gene body. We kept only bi-allelic SNPs that exhibited minor allele frequency 0.05 and HWE p-value $> 10^{-5}$ captured. We then fit elastic net and SuSiE models to the log-transformed and normalized gene expression data and restricted the analysis to genes with expression *cis*-heritability p-value $< 0.01$. To evaluate the performance of these approaches, we computed the out-of-sample $R^2$ using 265 newly added individuals in GTEx v8.

## Power analysis for ASE prediction

We performed power analysis to estimate the fraction of the cases that the current eQTL data fully described ASE signal. Statistical power facilitates the interpretation of the results, where the low read counts

or other features make it less likely to observe significant differences between the observed and predicted values. Factors that affect the power of the statistical test are the amount of total count, reference ratio, and the minimum fold change between the predicted and observed value.

For this purpose, we simulated the hypothesis ($H_1$) in which the log(aFC) of data is systematically off from the null hypothesis ($H_0$) by specific fold change ($fc$) (Eq. (5)), assuming $H_0$ data is binomial distributed with null reference ratio $r_O$

$$\log(aFC)_{H1} = \log(aFC)_{H0} \pm fc \qquad (5)$$

The log(aFC) is defined as the logit function of the reference ratio ($r_i$) (Eq. (6)).

$$\log(aFC)_{Hi} = \log\left(\frac{r_i}{1 - r_i}\right) \qquad (6)$$

One thousand binomial samples were produced from the hypothesis $H_1$ (500 samples from $\log(aFC)_{H1} = \log(aFC)_{H0} + fc$, and 500 samples from $\log(aFC)_{H1} = \log(aFC)_{H0} - fc$) for different levels of read coverage and reference ratios. The significant difference between the generated samples of $H_1$ and the null hypothesis $H_O$ was determined by the binomial test (nominal $p$ value < 0.01), where each gene is tested against its own null model described by reference ratio and read counts.

Power estimation based on simulation for different amount of read counts and reference ratios considering specific fold changes ($\Delta aFC$) 0.5 and 1 were calculated (Supplementary Fig. 10A). Here is an example to give an idea of how the fold resolution affects the reference ratios. Assuming the null hypothesis $\log_2$(aFC) to be zero, the $r_0$ is 0.5, the fold resolution of 0.5 and 1 corresponds to reference ratios of about 0.59 (or 0.41) and 0.67 (or 0.33), respectively (Supplementary Fig. 10B).

Applying power statistics analysis on the GTEx v8 haplotype-aggregated ASE data, for each individual, the protein-coding genes with TPM > 1 available at different levels of power (for 0.5- and 1-fold resolution) were considered for downstream analysis (Supplementary Fig. 10C). To obtain the fraction of genes available at each level of power, the significance of $\Delta aFC$ (difference of aFCs between the predicted and observed values), was determined by the binomial test at a 1% $p$ value threshold with the cut-off of 0.5 (for 0.5-fold resolution) or 1 (for onefold resolution) for $\Delta aFC$ to avoid false positives in high expressed genes.

### Ancestry-specific aFC estimation

We generalized the aFC-n model to calculate ancestry-specific aFC for European-American and African-American sub-populations using (Eq. (7)),

$$\log_2 e_{<i_1 \ldots i_N>, <j_1 \ldots j_N>} = \log_2\left(2^{(s_E(1-I) + s_A I).h1} + 2^{(s_E(1-I) + s_A I).h2}\right) + \log_2 e_R \qquad (7)$$

where $h_1$ is a binary vector representing the phased genotype of each allele and $I$ is an indicator of ancestry background for each individual with a fixed value (1 for African-Americans and 0 otherwise). The estimates $s_E$ and $s_A$ represent the aFC for European and African populations, respectively. The ancestry-specific aFC was selected for cases with non-overlapping confidence intervals and for the rest of the cases we used effect size estimates derived from the standard aFC-n model.

### $V^G$ estimation

$V^G$ estimates were calculated over GTEx v8 samples by the analysis of expression variation (ANEVA)[15]. We analyzed genes with at least 30 reads in at least 6 donors and at least 5000 reads in all individuals in a target tissue. The median of $SD^G$ ($\sqrt{V^G}$) was calculated for tested genes for each sample and the median of the resulting values was calculated across tissues (Fig. 4F).

### Reporting summary

Further information on research design is available in the Nature Portfolio Reporting Summary linked to this article.

## Data availability

All GTEx protected data are available through the database of Genotypes and Phenotypes (dbGaP) (accession no. phs000424.v8). Public-access data, including conditionally independent eQTLs, gene read counts and haplotype expression matrix, are available on the GTEx Portal (http://gtexportal.org/) as downloadable files.

The aFC estimates generated in this study that support the effect sizes for all independent eQTLs in GTEx v8 have been deposited in the Zenodo database, publicly accessible on https://doi.org/10.5281/zenodo.10002703. This data is also available on GTEx portal as a downloadable file.

## Code availability

Software for calculating aFC from independent eQTL data is available online here: https://doi.org/10.5281/zenodo.8412460 and also on https://github.com/PejLab/aFCn.

Software for calculating predicted ASE and gene expression using allelic fold change, as well as simulated ASE and gene expression, employing simulated allelic fold change are available online here: https://doi.org/10.5281/zenodo.8409098 and also on https://github.com/PejLab/gene_expr_pred.

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

## Acknowledgements

We thank the GTEx donors for their contributions to science, the GTEx Laboratory, Data Analysis, and Coordinating Center (LDACC), and the GTEx analysis working group (AWG) for their work in generating the resource. N.E, N.M and P.M were supported by NIGMS award R01GM140287. N.E and P.M were supported by a collaborative research agreement with Takeda California, Inc. P.M. was partly supported by Skaggs Scholars Program. S.E.C. was supported by NHGRI grant 1K99HG009916-01.

## Author contributions

N.E and P.M conceived the work and wrote the manuscript. B.M.K implemented the aFC pipeline. E.J.S calculated the genetic variation in gene expression estimates. N.M performed the SuSiE and elastic net analysis. S.E.C generated the haplotype-aggregated allelic expression data. N.E performed all other analyses. All the authors provided critical feedback on the manuscript.

## Competing interests

S.E.C. is a co-founder, Chief Technology Officer, and stock owner at Variant Bio. The remaining authors declare no competing interests.
