## [Peer Review File · Nature Communications]

Haplotype-aware modeling of cis-regulatory effects highlights the gaps remaining in eQTL dataREVIEWER COMMENTS

Reviewer #1 (Remarks to the Author):

This study presents a multi-variant approach to accurate estimation of the cis-regulatory effect for conditionally independent eQTLs associated with a gene. The approach extends an earlier approach developed by the senior author. The authors show that the accuracy gap between the new approach and the earlier approach widened in genes with more known eQTLs, with predictions significantly more accurate for multi-eQTL genes. The new method also improves prediction of the so-called genetically regulated expression (GR_EX), which has been used to identify expression-mediated associations in TWAS. The study is significant, but I have some concerns with the paper as it currently stands. The following must be addressed to improve the interpretability of the results:

1. It is not clear why the elastic net predictive model was performed on a filtered set of SNPs (i.e., the conditionally independent eQTLs [lines 125-126]). This would imply that the minimization of the elastic net loss function (and regularized terms) was done on a subset of the SNPs, possibly leading to models that are not 'optimal' for the cis region of a gene.

2. Much of the analyses (represented in figures 1-3) were done only in adipose tissue. Is there any reason for this? How would the results vary with tissue? Some tissues have high cell type heterogeneity; others less so.

3. Please justify the use of binomial and poisson noise in the simulation scheme (for haplotype counts and expression).

4. Line 293: Why not use R instead of R² to assess prediction accuracy? It's possible that the predicted value is negatively correlated with the observed (as others have noted, e.g., Zhou et al. Nat Gen 2020).

5. The absence of a summary-statistics based approach to identify trait associations with genetically driven gene dosage limits the application of the proposed method to existing GWAS datasets. What are the challenges in developing such an approach? What will be required?

Reviewer #2 (Remarks to the Author):

This is a paper that extends the authors' original research of 2017 on aFC from one eQTL (or aFC_1) to multiple independent eQTLs (or aFC_n). The methodological contribution is rather limited as the idea has been already given in their original paper and the estimation procedure is also simple and straightforward. The major strength of the paper is on its software development. The paper in general is well written with some major and minor issues summarized below:

1. Is the conditional independence assumption among eQTLs necessary for the proposed procedure? If yes, how does the selection procedure impact the downstream aFC_n analysis?
2. In your method comparison, different procedures and eQTLs are used for aFC_n, SuSiE and elastic net analysis, which raises a serious concern. For example, age is only included as a confounder for SuSiE and elastic net analysis but not for aFC_n. Also aFC_n only uses conditional independent eQTLs while the other two procedures use all the eQTLs that meet their filtering criteria.
3. In Fig. 1G, does it make any sense to have negative R2s? How was the described variance (R2) calculated exactly?

Also what is n in Fig 1 legends?

4. In contrast to traditional eQTL analysis where FDR or conservative p-value cutoff is commonly used for controlling of multiple testing, the paper uses very liberal p-values. Any explanations for doing this?
5. line 316 of page 18: If model accuracy is evaluated on testing samples, why would over-fitting be a concern? Furthermore, with no provided details on how the data is actually permuted, it is impossible to judge if the proposed permutation is valid or not.

Reviewer #3 (Remarks to the Author):

In their manuscript, "Haplotype-aware modeling of cis-regulatory effects highlights the gaps remaining in eQTL data", Ehsan and colleagues introduce aFC-n, a method for estimating cis-regulatory effects in genes with multiple independent eQTLs using a multi-variant generalization of allelic fold change (aFC-1, a concept they introduced in previous work). They first apply aFC-n to simulated expression data and show that it produces accurate eQTL effect sizes estimates (as quantified by allelic fold change). Next, they applied aFC-n to data from the Genotype-Tissue Expression (GTEx) v6 project to jointly estimate eQTL effect sizes for variants with conditionally independent effects on gene expression (as identified by stepwise regression) and use these estimates to predict the genetically regulated gene expression of v8 GTEx samples (held-out set). They then compare the prediction performance of their method aFC-n to the performance of the aFC-1, elastic net, and SuSiE methods. They show that predictors based on eQTL effects estimated from aFC-n show higher prediction accuracy than predictors from aFC-1 (for genes with more than one conditionally independent eQTL), and the elastic net and SuSiE methods. Finally, for each

individual, they compare the observed allelic imbalance to the allelic imbalance predicted by the eQTL effects estimated from aFC-n and show that, while a large percentage of the allelic observed imbalance is explained by eQTLs (22% decrease in nr of genes with imbalance after accounting for eQTL effects), 77% of genes still show a residual imbalance of more than .5 log₂ fold or more after accounting for the eQTL effects. The manuscript is, for the most part, clearly written and most statistical analyses are valid. However, I find their comparison to existing methods for predicting cis-genetically regulated gene expression very problematic (see major points below).

Major points

1. Accuracy of effect size prediction. The authors use simulated data to show that aFC-n provides accurate estimates the eQTL effect sizes when all causal variants are known (Page 1: line 12, page 4-5: line 71-72, & Sup Fig 2). However, this will not always be the case in real data applications. Specifically, the estimates will be accurate (i.e., unbiased) only if (1) all eQTL variants of a gene are independent (not in LD with each other) or (2) if eQTL variants are correlated but all eQTLs for a gene are known and included in the aFC-n model. The latter is not guaranteed in practice, especially when stepwise regression is used to identify all relevant variants for a gene. Ideally, they should further explore the impact of stepwise regression on the accuracy of their effect size estimation under different LD structures, e.g., via simulations. At minimum, they should update the text in the abstract 'to enable more accurate quantification' and on page 5, line 72 "aFC-n consistently estimated the effect size accurately across all genes, when all eQTL variants of a gene were included in the model". They should also elaborate on this limitation in the Discussion section.

2. Improvement in accuracy of effect size prediction over existing tools. In the abstract, the authors claim that aFC-n significantly improves the accuracy to estimate eQTL effect size over the current tools. I am not sure what current tools they are referring to, is that aFC-1? In addition, nowhere in the manuscript I could find a comparison of effect size estimation accuracy between different methods. If they are not showing this, they should remove this claim from the manuscript. I have added some suggestions in the minor comments (#1) if the authors choose to do a comparison with existing tools.

3. Comparison to elastic net and SuSiE models. The authors compare the gene expression prediction accuracy of aFC-n to the accuracy of elastic net and SuSiE methods. For predicting expression using the aFC model they used independent eQTLs derived from GTEx v6p data using stepwise regression. However, for the other two models, they used all genetic variants in the 1Mb window around each gene. They show that predictors based on eQTL effects estimated from aFC-n show higher prediction accuracy than predictors from the elastic net and SuSiE methods. I find this comparison particularly problematic: the other two methods are expected to have poorer prediction performance than a model with one or few predictors (aFC-n) as they must simultaneously perform variable selection. The authors need to compare the performance of these three methods on the exact same number of SNPs before drawing any conclusions about the superiority of their method. Since they cannot fit their method on all SNPs

within 1Mb of a gene, they should at least fit elastic net (or SuSiE) on the same set of variants they use for aFC-n. Otherwise, they should drop that section from their paper.

Minor comments

1. The manuscript would benefit from comparing the effect size estimates from aFC-1 and aFC-n when multiple eQTLs exist for a gene and the eQTL variants are (highly) correlated since that is the scenario when one would benefit from jointly estimating the effects of multiple variants. Specifically, when eQTLs are independent (not in LD) aFC-1 would return unbiased estimates while aFC-n will return both unbiased and more efficient estimates (i.e., effect sizes will have smaller standard errors than those estimated by aFC-1). When eQTLs are correlated, aFC-1 would return biased estimates while aFC-n will return unbiased estimates. The fact that aFC-1 (and any method that estimates effects separately for each variant) will lead to biased estimates when multiple *correlated* eQTLs exist for a gene might also partly explain why the correlation between aFC-1 and aFC-n estimates drops when moving from genes with one eQTL to genes with multiple conditionally independent eQTLs (Figure 1B-C). It seems like the 'conditionally independent' eQTLs identified in GTEx might be in LD. It would be helpful for the reader to mention that when discussing these results in page 5, lines 76-78: "The correlation ranges from 89%-98% across tissues for genes with a single eQTL where the two methods are mathematically identical, and from 80%-92% for genes with multiple eQTLs (Fig.1B,C), possibly as a result of biased aFC-1 estimates when eQTL variants are in LD". Moreover, the authors used aFC estimates derived from the aFC-1 and aFC-n models to predict gene expression due to cis-genetic effects and show that aFC estimates from the aFC-n model led to more accurate prediction of gene expression (Figure 1D&F). While this is a valid observation, the authors offer no explanation of where this improvement is coming from. Elaborating a bit on the reasons behind this discrepancy can give insight both on the types of (conditionally independent) eQTLs found in GTEx and scenarios when aFC-n is useful. I believe this is a result of eQTL variants within a gene being in LD which leads to biased effect size estimates when using aFC-1.

2. The manuscript would benefit from some further justification of why having an estimated (multi variant) allelic fold change is a useful metric besides the one line in page 2 lines 33-34, e.g., could it provide a more powerful test for identifying the effect of rare eQTLs?

3. What does the y-axis in Figure 3 show? Is this the average prediction accuracy across all genes in that set? It seems from Figure 3 that the increase in the performance of aFC-n over SuSiE is less than 1%. Could you report in the main text the exact difference in accuracy between methods with accompanying p-values?

Response letter, Ehsan et al. (NCOMMS-22-03940-T)

The reviewer questions are in black while our answers are in blue. In this response letter, the text quoted from the revised manuscript is designated in *italic*

REVIEWER COMMENTS

Reviewer #1 (Remarks to the Author):

This study presents a multi-variant approach to accurate estimation of the cis-regulatory effect for conditionally independent eQTLs associated with a gene. The approach extends an earlier approach developed by the senior author. The authors show that the accuracy gap between the new approach and the earlier approach widened in genes with more known eQTLs, with predictions significantly more accurate for multi-eQTL genes. The new method also improves prediction of the so-called genetically regulated expression (GR_EX), which has been used to identify expression-mediated associations in TWAS. The study is significant, but I have some concerns with the paper as it currently stands. The following must be addressed to improve the interpretability of the results:

The authors would like to thank the reviewer for constructive remarks and positive consideration of the paper.

1. It is not clear why the elastic net predictive model was performed on a filtered set of SNPs (i.e., the conditionally independent eQTLs [lines 125-126]). This would imply that the minimization of the elastic net loss function (and regularized terms) was done on a subset of the SNPs, possibly leading to models that are not 'optimal' for the cis region of a gene.

We thank the reviewer for raising this point and apologize for the lack of clarity in the original text. The analysis in the original manuscript was indeed done on *all SNPs* within a 1Mb window for each gene for elastic net and SuSiE, in line with the standard TWAS practice.

Incidentally, another reviewer has requested us to assess the performance of elastic net and SuSiE when applied to *the same set of SNPs* used by aFC-n. Therefore, in the revised manuscript, we now compare genetically regulated gene expression prediction between the three models (aFC-n, SuSiE and elastic net), under two distinct scenarios:

- 1) when all genetic variants in the 1Mb window around each gene are included in SuSiE and elastic net (Standard TWAS practice), with and without correcting for confounding factors (Fig. 3; Supplementary Fig. 8A-B).
- 2) when only conditionally independent eQTL SNPs for a gene are included in the model (identical set of variants used for all three methods), with adjusted expression counts (Supplementary Fig. 8C-D).

The aFC-n performance is higher than the other two methods under both scenarios. We updated Fig. 3 and added Supplementary Fig. 8 [See below], and revised the accompanying text description as follows (subsection: The aFC-n improves the prediction of genetically regulated gene expression):

“We used elastic net (*enet*)²⁶ and Sum-of-Single-Effects (*SuSiE*)²⁷, two powerful and robust methods used for predicting gene expression from genetic data to benchmark the accuracy of predicted gene expressions from eQTL effect sizes (Methods) and for that, we used 1) all genetic variants in the 1Mb window around each gene meeting our QC criteria (Fig.3, Supplementary Fig.8A-B), and 2) conditionally independent eQTLs for a gene (Supplementary Fig.8C-D). Restricting the comparison to genes with cis-heritability p -value <0.01 present in eQTL data, we found that the prediction performance of the eQTL genotypes in the aFC model was higher than the two state-of-the-art methods in unseen samples.”

Fig.3: The aFC-n improves predictive accuracy of genetically regulated gene expression.

Supplementary Fig.8: Performance of aFC-n, elastic net and SuSiE applied on adjusted gene expression, using all SNPs within 1Mb window for each gene (A-B) and using the same set of SNPs used by aFC-n (C-D).

2. Much of the analyses (represented in figures 1-3) were done only in adipose tissue. Is there any reason for this? How would the results vary with tissue? Some tissues have high cell type heterogeneity; others less so.

We used adipose subcutaneous tissue as one of the highly sampled tissues in GTEx with 581 samples. We now perform additional analysis and add new figures to show that aFC-n performs well

consistently across all GTEx tissues (Fig. 1H [See below]; Supplementary Fig. 4), and that the performance is not significantly associated with cell-type heterogeneity in different tissue samples (Supplementary Fig. 5 [See below]).

To evaluate how the results vary across tissues, we performed the comparison of aFC-n and aFC-1 in terms of coefficient of determination (R^2) for all tissues. We obtained a similar pattern for total expression prediction in 49 GTEx tissues. Fig. 1H is added to the manuscript to represent the variance explained by eQTLs in gene expression, aggregated across 49 GTEx tissues. In addition, Supplementary Fig. 4 shows the results for each individual tissue. We also included the distribution of absolute effect size and the number of eGenes across tissues in Supplementary Figures 3 and 11, respectively to improve the interpretability of the results across tissues

To assess the effect of cell type heterogeneity (CTH) on the aFC-n performance, we calculated the CTH for 49 GTEx tissues using xCell enrichment scores for 7 cell types (adipocytes, epithelial cells, hepatocytes, keratinocytes, myocytes, neurons, neutrophils) using the sum of standard deviation for cell enrichment scores. We did not observe a significant association between CTH and R^2 obtained from aFC-n model (Spearman correlation p-value 0.27). These results are shown in the new Supplementary Fig.5. Below are the newly added panels:

Fig. 1H) Variance explained by eQTLs in gene expression aggregated across 49 GTEx tissues.

Supplementary Fig. 5: We did not observe correlation between cell type heterogeneity and the median R^2 for each tissue.

3. Please justify the use of binomial and poisson noise in the simulation scheme (for haplotype counts and expression).

Thank you for raising this issue. We have updated the text in the methods section to clarify this. In our simulation, we used binomial and poisson distribution to draw discrete simulated read counts from the continuous expected reference ratios, and gene expressions, respectively. The simulation was intended for validation of the overall code implementation and inference procedure stability. For simplicity, we did not include any extra-Poisson biological noise in the simulation since the simulation statistics were not intended for evaluating the real-world performance of the tool which was done using the real GTEx data. The current text in the method section reads as follows:

“We used simulated data to validate aFC-n implementation and inference stability. We used 15,167 genes from adipose subcutaneous, their associated eQTLs (range: 1 to 14 eQTLs per gene), and individual genotypes for simulation. We synthesized log2 effect sizes for each eQTL from standard Normal distribution (norm[0,σ=1]). We calculated the expected haplotypic expression using Eq. 1 and used the empirical gene expression average for each gene to set e_R . Expected gene expression for each GTEx individual was estimated as the sum of the two haplotypic counts, and the simulated sequencing read count for each gene was generated using a Poisson distribution. To generate simulated ASE data, we calculated the expected reference expression ratio for each individual as $r = \frac{e_{h1}}{e_{h1} + e_{h2}}$, and simulated discrete read counts for each haplotype using Binomial distribution. Notably, we did not include any additional biological variation in our simulated data beyond genotypic differences among the individuals, and the sampling noise associated with the count nature of the data. This was to maintain simplicity since the simulations were intended for validating the code and inference procedure and not for reporting performance measures which we performed using real GTEx data.”

4. Line 293: Why not use R instead of R² to assess prediction accuracy? It's possible that the predicted value is negatively correlated with the observed (as others have noted, e.g., Zhou et al. Nat Gen 2020).

We opted for using the coefficient of determination, R², as it is a general measure, and it directly reports the fraction of variance described by the model. Notably, we calculate R² using a general definition as 1 – (residual variation/data variation). As such, our R² estimates can and do take negative values in certain scenarios, including when the predictions are anti-correlated with the data. We have modified the text to clarify the way R² is derived to describe the proportion of the variance that is explained by the model as follows:

(Methods: Imputing Gene and allelic expression):

“The prediction accuracy was measured by, R² (Eq. 3), between predicted and observed gene expression to describe the proportion of the variance that is explained by the model.”

$$R^2 = 1 - \frac{\text{Unexplained variation}}{\text{Data variation}} \quad (3)$$

5. The absence of a summary-statistics based approach to identify trait associations with genetically driven gene dosage limits the application of the proposed method to existing GWAS datasets. What are the challenges in developing such an approach? What will be required?

Thanks for raising an important issue. We have expanded the discussion to further emphasize the point that, unlike regression-based methods, our method cannot be used to analyze associations based on summary statistics but increasing access to individual-level data can address this bottleneck by providing essential data for researchers. Here is the updated text in the Discussion section of the manuscript:

“Unlike the conventional regression-based methods, predictions of haplotype-level dosage cannot be derived and used to analyze associations based on summary statistics. However, others have developed approximation methods that utilize aFC framework for gene dosage prediction and related association analyses using summary statistics³⁸. Increased availability of large-scale genomic datasets on cloud services such as the UKBB research analysis platform and the NHGRI genomic data science analysis, visualization, and informatics lab-space (AnVIL) will facilitate individual-level analysis of genetic data in the coming years.”

Reviewer #2 (Remarks to the Author):

This is a paper that extends the authors' original research of 2017 on aFC from one eQTL (or aFC_1) to multiple independent eQTLs (or aFC_n). The methodological contribution is rather limited as the idea has been already given in their original paper and the estimation procedure is also simple and straightforward. The major strength of the paper is on its software development. The paper in general is well written with some major and minor issues summarized below:

1. Is the conditional independence assumption among eQTLs necessary for the proposed procedure? If yes, how does the selection procedure impact the downstream aFC_n analysis?

We thank the reviewer for highlighting this point. Conditional independence assumption is critical for joint estimation of the effect sizes via aFC-n. We have added text to Methods and Discussion to clarify the independence issue among the SNPs used in the model.

We added the following explanation in the Methods section (Haplotypic aFC estimation):

“Conditional independence assumption used in stepwise regression approach for deriving eQTLs identifies independent signals for gene expression and chooses the variants that describe the best signal while controlling for covariates and all other mapped eQTL signals^{3,4}. This enables us to estimate the regulatory effects by ensuring stability and identifiability of the model parameters in the optimization process.”

We added the following explanation in the Discussion section:

“Furthermore, aFC-n assumes that eQTL genotypes are linearly independent which is a critical condition for mathematical identifiability of the effect sizes and as such inherently satisfied in conditional eQTL datasets. However, the general application of aFC-n model on an arbitrary set of SNPs will require the addition of appropriate shrinkage penalties to enable parameter inference. Moreover, aFC-n assumes biological independence among eQTLs in that it does not allow for epistatic interactions between eQTLs. While there are many biological scenarios under which two regulatory variants can have nonadditive effects, we have previously shown that this assumption is rarely violated for the eQTL SNPs identified by stepwise regression approach³.”

2. In your method comparison, different procedures and eQTLs are used for aFC_n, SuSiE and elastic net analysis, which raises a serious concern. For example, age is only included as a confounder for SuSiE and elastic net analysis but not for aFC_n. Also aFC_n only uses conditional independent eQTLs while the other two procedures use all the eQTLs that meet their filtering criteria.

This is a valid observation. To apply a fair comparison for predicting genetically regulated gene expression between the three models, in the revised manuscript, we now compare genetically regulated gene expression prediction between the three models (aFC-n, SuSiE and enet), under two distinct scenarios:

- 1) when all genetic variants in the 1Mb window around each gene are included in SuSiE and elastic net (Standard TWAS practice). We fit elastic net and SuSiE models to the log-transformed and normalized gene expression data a) without correcting for confounding factors (Fig.3), and b) adjusting for the top 5 genotyping PCs, sequencing protocol (PCR-based or -free), sequencing platform (Illumina HiSeq 2000 or HiSeq X) and sex (Supplementary Fig.8A-B).
- 2) when only conditionally independent eQTL SNPs for a gene are included in the model (identical set of variants used for all three methods), with adjusted expression counts, (Supplementary Fig. 8C-D).

The aFC-n performance is higher than the other two methods under both scenarios. We updated Fig. 3, added Supplementary Fig. 8 [See reviewer #1, comment #1] and revised the accompanying text description as follows (subsection: The aFC-n improves the prediction of genetically regulated gene expression):

“We used elastic net (enet)²⁶ and Sum-of-Single-Effects (SuSiE)²⁷, two powerful and robust methods used for predicting gene expression from genetic data to benchmark the accuracy of predicted gene expressions from eQTL effect sizes (Methods) and for that, we used 1) all genetic variants in the 1Mb window around each gene meeting our QC criteria (Fig.3, Supplementary Fig.8A-B), and 2) conditionally independent eQTLs for a gene (Supplementary Fig.8C-D). Restricting the comparison to genes with cis-heritability p-value <0.01 present in eQTL data, we found that the prediction performance of the eQTL genotypes in the aFC model was higher than the two state-of-the-art methods in unseen samples”

3. In Fig. 1G, does it make any sense to have negative R2s? How was the described variance (R2) calculated exactly?

Also what is n in Fig 1 legends?

The R^2 reported in Fig. 1G is the prediction accuracy of allelic imbalance applied to the haplotype-aggregated allelic expression. There are two points to clarify, first the R^2 is used to describe the proportion of the variance that is explained by the model and is computed as $1 - \frac{\text{Unexplained variation}}{\text{Data variation}}$. The R^2 estimates can result in negative values when residual variation exceeds the initial variation in the data. Second, in this evaluation, the R^2 is computed to benchmark the effect size estimation on allele-specific expression, and the trained model is agnostic to this test data. The explanation provided in the subsection (The aFC-n improves the accuracy of the cis-regulatory effect size estimates) is as follows:

“Since both methods are agnostic to allele-specific expression data, using allelic imbalance prediction accuracy allows us to evaluate the quality of the effect size estimates in an orthogonal way¹³. We compared the predicted gene expressions from each set of effect size estimates with observed gene expression after log-transformation and PEER correction³. The predicted allelic imbalance was benchmarked against the observed logit-transformed haplotype-aggregated allelic expression generated by phASER^{14,25}. “

We also edited the Methods section (Imputing Gene and allelic expression) to clarify the computation of R^2 :

“The prediction accuracy was measured by, R^2 (Eq. 3), between predicted and observed gene expression to describe the proportion of the variance that is explained by the model.”

$$R^2 = 1 - \frac{\text{Unexplained variation}}{\text{Data variation}} \quad (3)$$

N denotes the number of analyzed genes. This has been added to the Fig. 1 legend.

4. In contrast to traditional eQTL analysis where FDR or conservative p-value cutoff is commonly used for controlling of multiple testing, the paper uses very liberal p-values. Any explanations for doing this?

The FDR procedure in the eQTL analysis is used to limit false discoveries by controlling the Family-wise error rate. Here, we use the nominal p-value a) for the power analysis and b) for identifying the fraction of genes with excess allelic imbalance. Controlling for family-wise error rate is not appropriate and is not conventionally applied in power analysis (case a) where we aim to estimate the statistical power at each reference ratio and read count level. In estimating the fraction of genes with excess allelic imbalance (case b), the reported statistic, *the fraction of the tested genes* with excess allelic imbalance, does not increase with the number of tests under the null hypothesis, so correcting for multiple hypothesis testing is not warranted.

5. line 316 of page 18: If model accuracy is evaluated on testing samples, why would over-fitting be a concern? Furthermore, with no provided details on how the data is actually permuted, it is impossible to judge if the proposed permutation is valid or not.

Thanks for pointing this out. Fig.1F aims to compare aFC-n and aFC-1 using conditionally independent eQTLs available on GTEx v8 project data. To show that the outcome is not subject to overfitting the training data, we had to either apply a k-fold cross-validation analysis or train the model on a permuted data. Cross-validation analysis requires performing eQTL calling for each iteration which would be computationally intense and would also decrease the power in eQTL calling due to subsetting the sample size. Thus, we made a permuted data by shuffling the gene expressions over the individuals, which will detach the connection between genotype and gene expression data, and we trained the model based on the shuffled data. The failure of observing a similar performance from the shuffled data indicates that the increased accuracy of the aFC-n model over aFC-1 is not happening by chance and could be explained by the fact that the aFC-n model accounts for regulatory effects of all eQTL alleles simultaneously.

On the other hand, Fig.1G shows the prediction accuracy of allelic imbalance to evaluate the effect sizes in an orthogonal way since the trained model is agnostic to allele-specific expression. We added the following explanation to elaborate on this point (Methods: Imputing gene and allelic expression):

“With an increased number of eQTLs for a gene, we observed a steadily increasing gap for the prediction accuracy between aFC-n and aFC-1 in multi-eQTL genes (Fig. 1F). To examine that the outcome is not subject to over-fitting the training data, we made a permuted data by shuffling the gene expressions over the individuals, which will detach the connection between genotype and gene expression data. We trained the model based on the shuffled data. The output showed that although the decreasing pattern was held, we did not observe similar performance from the shuffled data. This means that the improvement in our results did not happen by chance and the increased accuracy could be explained by the fact that the aFC-n accounts for the regulatory effects of all eQTL alleles simultaneously and by doing so it accounts for phasing dependent effects and linkage disequilibrium (Fig. 1I). To further evaluate the effect sizes on an independent data we used haplotype-aggregated allelic expression since the trained model is agnostic to allele-specific expression (Fig 1G).”

Reviewer #3 (Remarks to the Author):

In their manuscript, “Haplotype-aware modeling of cis-regulatory effects highlights the gaps remaining in eQTL data”, Ehsan and colleagues introduce aFC-n, a method for estimating cis-regulatory effects in genes with multiple independent eQTLs using a multi-variant generalization of allelic fold change (aFC-1, a concept they introduced in previous work). They first apply aFC-n to simulated expression data and show that it produces accurate eQTL effect sizes estimates (as quantified by allelic fold change). Next, they applied aFC-n to data from the Genotype-Tissue Expression (GTEx) v6 project to jointly estimate eQTL effect sizes for variants with conditionally independent effects on gene expression (as identified by stepwise regression) and use these estimates to predict the genetically regulated gene expression of v8 GTEx samples (held-out set). They then

compare the prediction performance of their method aFC-n to the performance of the aFC-1, elastic net, and SuSiE methods.

They show that predictors based on eQTL effects estimated from aFC-n show higher prediction accuracy than predictors from aFC-1 (for genes with more than one conditionally independent eQTL), and the elastic net and SuSiE methods. Finally, for each individual, they compare the observed allelic imbalance to the allelic imbalance predicted by the eQTL effects estimated from aFC-n and show that, while a large percentage of the allelic observed imbalance is explained by eQTLs (22% decrease in nr of genes with imbalance after accounting for eQTL effects), 77% of genes still show a residual imbalance of more than .5 log₂ fold or more after accounting for the eQTL effects. The manuscript is, for the most part, clearly written and most statistical analyses are valid. However, I find their comparison to existing methods for predicting cis-genetically regulated gene expression very problematic (see major points below).

Major points

1. Accuracy of effect size prediction. The authors use simulated data to show that aFC-n provides accurate estimates the eQTL effect sizes when all causal variants are known (Page 1: line 12, page 4-5: line 71-72, & Sup Fig 2). However, this will not always be the case in real data applications. Specifically, the estimates will be accurate (i.e., unbiased) only if (1) all eQTL variants of a gene are independent (not in LD with each other) or (2) if eQTL variants are correlated but all eQTLs for a gene are known and included in the aFC-n model. The latter is not guaranteed in practice, especially when stepwise regression is used to identify all relevant variants for a gene. Ideally, they should further explore the impact of stepwise regression on the accuracy of their effect size estimation under different LD structures, e.g., via simulations. At minimum, they should update the text in the abstract ‘to enable more accurate quantification’ and on page 5, line 72 “aFC-n consistently estimated the effect size accurately across all genes, when all eQTL variants of a gene were included in the model”. They should also elaborate on this limitation in the Discussion section.

We thank the reviewer for highlighting this issue. We agree with the reviewer and have improved the text in the abstract, the main text, and the discussion as requested. Furthermore, we now include a new analysis to show the detrimental effect of LD contamination on effect size estimates. These modifications are detailed below.

Update text in Abstract:

“Here we introduce a multi-variant generalization of allelic Fold Change (aFC), aFC-n, to enable quantification of the cis-regulatory effects in multi-eQTL genes under the assumption that all eQTLs are known and conditionally independent.”

Update text in Page 5 (Subsection: The aFC-n improves the accuracy of the cis-regulatory effect size estimates):

“In this simulation study, aFC-n consistently estimated the effect size accurately across all genes, when all eQTL variants of a gene were included in the model.”

The following sentences were added to the Discussion:

“We showed that aFC-n provides accurate estimates of cis-regulatory effects when all regulatory variants affecting a gene are known and conditionally independent. Violation of these assumptions can affect the quality of the results. Specifically, we demonstrated how the LD between two eQTLs affecting the same gene can systematically erode the performance of the aFC-1. While this experiment demonstrates the strength of aFC-n by simultaneously estimating all multiple effect sizes, it also highlights its limitation in cases where one or more eQTLs are not included in the model. Specifically, when a gene is affected by other eQTL beyond what is included in the model, the effect size estimates will be systematically biased by the contaminating effect of the LD. Furthermore, aFC-n assumes that eQTL genotypes are linearly independent which is a critical condition for mathematical identifiability of the effect sizes and as such inherently satisfied in conditional eQTL datasets. However, the general application of aFC-n model on an arbitrary set of SNPs will require the addition of appropriate shrinkage penalties to enable parameter inference. Moreover, aFC-n assumes biologically independent among eQTLs in that it does not allow for epistatic interactions. While there are many biological scenarios under which two regulatory variants can have nonadditive effects, we have previously shown that this assumption is rarely violated for the eQTL SNPs identified by stepwise regression approach³.”

New analysis on the effect of LD on effect size estimates:

If an unknown regulatory variant is in LD with known eQTLs it will systematically bias the effect size estimates. To demonstrate this, we used genes with two eQTLs and compared aFC-1 and aFC-n in terms of the estimated effect sizes and the described variance using simulated and real data. The aFC-n and aFC-1 models are analytically identical in the case of single eQTLs. Thus, in this experiment, aFC-1 represents the limited eQTL knowledge scenario in which only one out of the two eQTL is known. The following figure panels were added:

Fig. 11) The prediction accuracy for genes associated with two independent eQTLs as a function of their linkage disequilibrium (LD). The SNPs are filtered to address eQTLs with $LD \geq 0.05$ (24,930 SNP-pair). The x-axis represents the median LD, among 30 equally sized bins and the y-axis represents the median R^2 for the genes within each bin.

Supplementary Fig. 6: The aFC-n model showed stable performance in effect size estimation and prediction accuracy on simulated data compared to aFC-1 at different levels of LD, for independent eQTL variants. A-B) The effect size estimation and prediction accuracy on simulated data. The x-axis represents different levels of LD between the eQTL variants for 2000 genes (200 genes at each level) associated with two eQTLs. The y-axis is the absolute difference between the simulated and predicted effect sizes (A) and prediction accuracy (B). C) The distribution of LD values among the eQTL variants of 2-eQTL genes in 49 GTEx tissues.

2. Improvement in accuracy of effect size prediction over existing tools. In the abstract, the authors claim that aFC-n significantly improves the accuracy to estimate eQTL effect size over the current tools. I am not sure what current tools they are referring to, is that aFC-1? In addition, nowhere in the manuscript I could find a comparison of effect size estimation accuracy between different methods. If they are not showing this, they should remove this claim from the manuscript. I have added some suggestions in the minor comments (#1) if the authors choose to do a comparison with existing tools.

The reviewer is correct that we only compare our results to those from aFC-1. We have updated the text throughout the manuscript to communicate this clearly. Additionally, we would like to thank the reviewer for the suggested analysis in minor comment #1. We now include this in the revised manuscript.

3. Comparison to elastic net and SuSiE models. The authors compare the gene expression prediction accuracy of aFC-n to the accuracy of elastic net and SuSiE methods. For predicting expression using the aFC model they used independent eQTLs derived from GTEx v6p data using stepwise regression. However, for the other two models, they used all genetic variants in the 1Mb window around each gene. They show that predictors based on eQTL effects estimated from aFC-n show higher prediction accuracy than predictors from the elastic net and SuSiE methods. I find this comparison particularly problematic: the other two methods are expected to have poorer prediction performance than a model with one or few predictors (aFC-n) as they must simultaneously perform variable selection. The authors need to compare the performance of these three methods on the exact same number of SNPs before drawing any conclusions about the superiority of their method. Since they cannot fit their method on all SNPs within 1Mb of a gene, they should at least fit elastic net (or SuSiE) on the same set of variants they use for aFC-n. Otherwise, they should drop that section from their paper.

We agree with the reviewer's point. Therefore, in the revised manuscript, we now compare genetically regulated gene expression prediction between the three models (aFC-n, SuSiE and elastic net), under two distinct scenarios:

- 1) when all genetic variants in the 1Mb window around each gene are included in SuSiE and elastic net (Standard TWAS practice), with and without correcting for confounding factors (Fig. 3A-B; Supplementary Fig. 8A-B).
- 2) when only conditionally independent eQTL SNPs for a gene are included in the model (identical set of variants used for all three methods), with adjusted expression counts (Supplementary Fig. 8C-D).

The aFC-n performance is higher than the other two methods under both scenarios. We updated Fig. 3, added Supplementary Fig. 8 [See reviewer#1, comment#1], and revised the accompanying text description as follows (subsection: The aFC-n improves the prediction of genetically regulated gene expression):

“We used elastic net (enet)²⁶ and Sum-of-Single-Effects (SuSiE)²⁷, two powerful and robust methods used for predicting gene expression from genetic data to benchmark the accuracy of predicted gene expressions from eQTL effect sizes (Methods) and for that, we used 1) all genetic variants in the 1Mb window around each gene meeting our QC criteria (Fig.3, Supplementary Fig.8A-B), and 2) conditionally independent eQTLs for a gene (Supplementary Fig.8C-D). Restricting the comparison to genes with cis-heritability p-value <0.01 present in eQTL data, we found that the prediction performance of the eQTL genotypes in the aFC model was higher than the two state-of-the-art methods in unseen samples.”

Minor comments

1. The manuscript would benefit from comparing the effect size estimates from aFC-1 and aFC-n when multiple eQTLs exist for a gene and the eQTL variants are (highly) correlated since that is the scenario when one would benefit from jointly estimating the effects of multiple variants. Specifically, when eQTLs are independent (not in LD) aFC-1 would return unbiased estimates while aFC-n will return both unbiased and more efficient estimates (i.e., effect sizes will have smaller standard errors than those estimated by aFC-1). When eQTLs are correlated, aFC-1 would return biased estimates while aFC-n will return unbiased estimates. The fact that aFC-1 (and any method that estimates effects separately for each variant) will lead to biased estimates when multiple *correlated* eQTLs exist for a gene might also partly explain why the correlation between aFC-1 and aFC-n estimates drops when moving from genes with one eQTL to genes with multiple conditionally independent eQTLs (Figure1B-C). It seems like the 'conditionally independent' eQTLs identified in GTEx might be in LD. It would be helpful for the reader to mention that when discussing these results in page 5, lines 76-78: “The correlation ranges from 89%-98% across tissues for genes with a single eQTL where the two methods are mathematically identical, and from 80%-92% for genes with multiple eQTLs (Fig.1B,C), possibly as a result of biased aFC-1 estimates when eQTL variants are in LD”. Moreover, the authors used aFC estimates derived from the aFC-1 and aFC-n models to predict gene expression due to cis-genetic effects and show that aFC estimates from the aFC-n model led to more

accurate prediction of gene expression (Figure 1D&F). While this is a valid observation, the authors offer no explanation of where this improvement is coming from. Elaborating a bit on the reasons behind this discrepancy can give insight both on the types of (conditionally independent) eQTLs found in GTEx and scenarios when aFC-n is useful. I believe this is a result of eQTL variants within a gene being in LD which leads to biased effect size estimates when using aFC-1.

We thank the reviewer for suggesting this analysis. As explained in major point #1, we now include a new analysis to show the detrimental effect of LD contamination on effect size estimates. We compared the aFC estimates from aFC-1 and aFC-n under different LD structures by simulating 2-eQTL genes with LD ranging from 0 to 0.9 and we showed the results in the Supplementary Fig. 6A-B [see Major points#1]. Moreover, we added Fig. 1I and Supplementary Fig. 6C [see Major points#1] to compare aFC-1 and aFC-n prediction accuracy for 2-eQTL genes on GTEx data as a function of LD calculated for conditional eQTLs associated with each gene. The description is updated in the manuscript as follows (subsection: The aFC-n improves the accuracy of the cis-regulatory effect size estimates):

“Different conditionally independent eQTL SNPs for a gene are regularly in linkage disequilibrium (LD). To further explore the effect of linkage disequilibrium (LD) on the performance of the aFC-1 and aFC-n, we simulated effect sizes for 2-eQTL genes with LD ranging from 0 to 0.9 (Methods). When eQTL variants within a gene are correlated, aFC-1 would return biased estimates which leads to a lower performance for effect size estimation and prediction accuracy in higher LD, while aFC-n showed consistent performance at all LD values (Supplementary Fig.6A-B). Moreover, for 2-eQTL genes in GTEx data, the prediction accuracy gap between aFC-n and aFC-1 was widened as eQTLs of a gene are in higher LD (Fig.1I; Supplementary Fig.6C), indicating that considering the regulatory effects of all eQTLs simultaneously, as is done in aFC-n, is critical for accurate estimation of regulatory effect in presence of linkage disequilibrium between eQTLs.”

2. The manuscript would benefit from some further justification of why having an estimated (multi variant) allelic fold change is a useful metric besides the one line in page 2 lines 33-34, e.g., could it provide a more powerful test for identifying the effect of rare eQTLs?

We added the following sentences to the Introduction to highlight this point.

“The aFC estimates quantify genetic effects on gene expression in an intuitive way and that is consistent with effect sizes from other assays such as allele-specific expression analysis, differential expression analysis, and RT-qPCR, and is mechanistically consistent with cis-regulation. Besides biological interpretability, aFC estimates have several mathematically convenient properties that facilitate downstream analysis, and as such, are used in a wide range of applications^{3,4,11,14-19}.”

3. What does the y-axis in Figure 3 show? Is this the average prediction accuracy across all genes in that set? It seems from Figure 3 that the increase in the performance of aFC-n over SuSiE is less than 1%. Could you report in the main text the exact difference in accuracy between methods with accompanying p-values?

In Figure 3A, the y-axis represents the median prediction accuracy (R^2) across all genes and the error bars represent 95% bootstrap confidence intervals of the median. Although for the shared genes, the absolute difference between the medians of aFC-n and SuSiE is about 0.01, the Wilcoxon signed rank test shows significant improvement for aFC-n. We added Figure 3B [See below] to show the median R^2 of SuSiE and elastic net relative to the median R^2 of the aFC-n model.

“Fig.3 | The aFC-n improves the predictive accuracy of genetically regulated gene expression. Comparing predicted gene expression from aFC-n, elastic net (enet)²⁶ and SuSiE²⁷ using out of sample data. For predicting gene expression from genetic data with SuSiE and enet we used all genetic variants in the 1Mb window around each gene meeting our QC criteria. A) The aFC-n model outperforms SuSiE and elastic net for 1,923 genes shared by all models in adipose subcutaneous tissue (median R^2 is 0.073 for the aFC-n model and 0.063 for the SuSiE predictive model; Wilcoxon signed rank test p-value: 1.8×10^{-47}). For the genes not shared between models, the performance is negligible for all models. Error bars represent 95% bootstrap confidence intervals of the median. B) The distribution of the median R^2 relative to the median of the R^2 for aFC-n model for 47 tissues. Comparing aFC-n with SuSiE and elastic net prediction models, Wilcoxon signed rank tests are significant (FDR < 0.05) for 46 and 47 tissues, respectively. The p-value annotation: ****: $p \leq 10^{-4}$.”

REVIEWER COMMENTS

Reviewer #1 (Remarks to the Author):

The authors have completely addressed my previous concerns.

Reviewer #2 (Remarks to the Author):

In the resubmission, the authors have addressed most of my previous concerns. However, there are still several serious concerns in the revision:

1. In Fig 3, it is unclear, how the genes associated with each model were identified. Are those the genes with significant eQTLs?

2. The authors have barely addressed my concern on the conditional independence assumption of the method "Is the conditional independence assumption among eQTLs necessary for the proposed procedure? If yes, how does the selection procedure impact the downstream aFC_n analysis?". They have only confirmed that the conditional independence assumption is critical, but have not provide a general guideline on how to select conditionally independent SNPs/eQTLs and how the selection procedure impacts the performance of aFC_n.

3. In equation (3), unexplained variation and data variation need to be defined

4. It is still unclear to me why "The failure of observing a similar performance from the shuffled data indicates that the increased accuracy of the aFC_n model over aFC₁ is not

happening by chance and could be explained by the fact that the aFC_n model accounts for regulatory effects of all eQTL alleles simultaneously." (see rebuttal letter page 8 lines 11-14)

5. Reviewer 1 requested the authors to "justify the use of binomial and poisson noise in the simulation scheme (for haplotype counts" (see rebuttal letter page 4 Q3), but clearly the authors have not addressed the question well.

Reviewer #3 (Remarks to the Author):

The authors have done an excellent job of revising their manuscript. All of my earlier concerns have been addressed and I have no other issues to raise.

Response letter, Ehsan et al. (NCOMMS-22-03940A)

The reviewer questions are in black while our answers are in blue. In this response letter, the text quoted from the revised manuscript is designated in *italics*.

Reviewer #1 (Remarks to the Author):

The authors have completely addressed my previous concerns.

Thank you for taking the time to review our manuscript and providing valuable feedback.

Reviewer #2 (Remarks to the Author):

In the resubmission, the authors have addressed most of my previous concerns. However, there are still several serious concerns in the revision:

1. In Fig 3, it is unclear, how the genes associated with each model were identified. Are those the genes with significant eQTLs?

Thank you for raising this issue. We have improved the figure caption to clarify this. Briefly, the genes associated with each model are identified according to their respective standard practices. For aFC-n we consider genes with at least one significant eQTL and for SuSiE/Elastic net we use genes with significant cis-h2. The figure 3A comparison includes three separate set of bars for genes that are specific to either or shared in both approaches. In figure 3B we only consider genes that are shared across models. These inclusion criteria are also detailed in the methods section: "*Mapping conditionally independent eQTLs in GTEx v6p data*", and "*Benchmarking gene expression prediction models*".

2. The authors have barely addressed my concern on the conditional independence assumption of the method "Is the conditional independence assumption among eQTLs necessary for the proposed procedure? If yes, how does the selection procedure impact the downstream aFC_n analysis?". They have only confirmed that the conditional independence assumption is critic, but have not provide a general guideline on how to select conditionally independent SNPs/eQTLs and how the selection procedure impacts the performance of aFC_n.

Thank you for the comment. Briefly, in the current manuscript we a) describe how conditionally independent eQTLs are mapped (methods section "*Mapping conditionally independent eQTLs in GTEx v6p data*"), b) provide analysis on impact of LD/missing eQTLs on the model performance (Supp. fig. 6), and c) provide text in discussion to cover surrounding issues.

Conditionally independent eQTLs are identified recursively by conditioning the analysis on the identified eQTLs for a given gene. This approach was used in the GTEx Consortium 2017 flagship paper, and it was later optimized and implemented in the TensorQTL tool (`--mode cis_independent`) which was published and used in the final GTEx release in 2020. We use conditionally independent eQTLs released by GTEx v8 throughout the paper, except for out of sample benchmarking where we had to map conditionally independent in GTEx v6 release. In this case we describe our used parameters in the section “*Mapping conditionally independent eQTLs in GTEx v6p data*”. While other variable selection approaches beyond stepwise regression used in TensorQTL are theoretically conceivable, we believe exploring these is out of the scope of this work.

This approach identifies SNPs that are 1) not linearly dependent, and 2) have significant individual effects. The latter is not a concern; if a SNP with individual effect is added to the model its effect will be estimated at zero. However, the former is a fundamental assumption in any regression model. In light of the analysis exploring the effect of LD presented in Supp. Fig. 6, we provide the following text in the discussion to provide further insight into the issue:

“We showed that aFC-n provides accurate estimates of cis-regulatory effects when all regulatory variants affecting a gene are known and conditionally independent. Violation of these assumptions can affect the quality of the results. Specifically, we demonstrated how the LD between two eQTLs affecting the same gene can systematically erode the performance of the aFC-1. While this experiment demonstrates the strength of aFC-n by simultaneously estimating all multiple effect sizes, it also highlights its limitation in cases where one or more eQTLs are not included in the model. Specifically, when a gene is affected by other eQTL beyond what is included in the model, the effect size estimates will be systematically biased by the contaminating effect of the LD. Furthermore, aFC-n assumes that eQTL genotypes are linearly independent which is a critical condition for mathematical identifiability of the effect sizes and as such inherently satisfied in conditional eQTL datasets. However, the general application of aFC-n model on an arbitrary set of SNPs will require the addition of appropriate shrinkage penalties to enable parameter inference. Moreover, aFC-n assumes biologically independent among eQTLs in that it does not allow for epistatic interactions. While there are many biological scenarios under which two regulatory variants can have nonadditive effects, we have previously shown that this assumption is rarely violated for the eQTL SNPs identified by stepwise regression approach³.”

3. In equation (3), unexplained variation and data variation need to be defined

Thank you for raising this concern. We have improved the text to address this issue. The current text reads as below (The equation number has changed to 4):

“The prediction accuracy for predicting gene expression log fold-change, and log-transformed allelic imbalance was measured by coefficient of determination, R^2 :

$$R^2 = 1 - \frac{\sum_{\text{individuals}}(\text{observation} - \text{prediction})^2}{\sum_{\text{individuals}}(\text{observation})^2} \quad (4)$$

where the prediction values are $\log_2 e_{\langle i_1 \dots i_N \rangle, \langle j_1 \dots j_N \rangle}$ for gene expression, and $AI_{\langle i_1 \dots i_N \rangle, \langle j_1 \dots j_N \rangle}$ for allelic imbalance as provided in Eq. 3, and the observed values are what is measured for gene expression fold change from mean, and log-transformed allelic imbalance measured for each individual in GTEx data.”

4. It is still unclear to me why "The failure of observing a similar performance from the shuffled data indicates that the increased accuracy of the aFC-n model over aFC-1 is not happening by chance and could be explained by the fact that the aFC-n model accounts for regulatory effects of all eQTL alleles simultaneously." (see rebuttal letter page 8 lines 11-14)

Thank you for your comment. The aFC-1 and aFC-n methods are mathematically identical except for the fact that aFC-n considers all eQTLs associated with a gene at once when estimating the effect sizes. We used permutation to ensure that the superior performance of aFC-n in gene expression prediction is not driven by overfitting. We have modified the sentence as follows to resolve the ambiguity:

“We used a permutation test to ensure that the improved performance of aFC-n versus aFC-1 is not driven by overfitting. Specifically, we permuted the individual sample IDs to decouple genotype and gene expression variation while retaining the data size, allele frequencies and LD structure. We found that the R^2 for the aFC-n predictions of gene expression in the permuted dataset remains at zero ruling out systematic overfitting to the data.”

5. Reviewer 1 requested the authors to "justify the use of binomial and poisson noise in the simulation scheme (for haplotype counts" (see rebuttal letter page 4 Q3), but clearly the authors have not addressed the question well.

We believe that this issue has been addressed as also indicated by the reviewer #1. We used binomial and Poisson distributions to draw discrete simulated read counts from the expected reference ratios, and gene expressions, respectively, which are standard choices for generating these count data. Also, we justify why we opted not to consider more sophisticated extra-binomial, or extra-poisson noise models in our simulations. Specifically, the simulation data was intended for validation of the overall code implementation and inference procedure stability, and we use real data to benchmark the performance of the model. Thus, the choice of the distributions used are not critical as they do not affect conclusions of the work.

Reviewer #3 (Remarks to the Author):

The authors have done an excellent job of revising their manuscript. All of my earlier concerns have been addressed and I have no other issues to raise.

Thank you for taking the time to review our manuscript and providing valuable feedback.

REVIEWERS' COMMENTS

Reviewer #2 (Remarks to the Author):

After multiple rounds of revisions, I'm happy to report that the paper has addressed all my concerns.